# Multi-fidelity Bayesian Optimization with Multiple Information Sources of Input-dependent Fidelity

**Mingzhou Fan**[1] **Byung-Jun Yoon**[1,2] **Edward Dougherty**[1] **Nathan Urban**[2]

**Francis Alexander**[3] **Raymundo Arróyave**[4,5] **Xiaoning Qian**[1,2,6]

[1]Department of Electrical & Computer Engineering, Texas A&M University, College Station, Texas, USA,
[2]Computational Science Initiative, Brookhaven National Laboratory, Upton, New York, USA,
[3]Argonne National Laboratory, Lemont, Illinois, USA,
[4]Department of Materials Science & Engineering, Texas A&M University, College Station, Texas, USA
[5]Department of Mechanical Engineering, Texas A&M University, College Station, Texas, USA
[6]Department of Computer Science & Engineering, Texas A&M University, College Station, Texas, USA

## Abstract

By querying approximate surrogate models of different fidelity as available information sources, Multi-Fidelity Bayesian Optimization (MFBO) aims at optimizing unknown functions that are costly if not infeasible to evaluate. Existing MFBO methods often assume that approximate surrogates have consistently high/low fidelity across the input domain. However, approximate evaluations from the same surrogate can have different fidelity at different input regions due to data availability and model constraints, especially when considering machine learning surrogates. In this work, we investigate MFBO when multi-fidelity approximations have input-dependent fidelity. By explicitly capturing input dependency for multi-fidelity queries in Gaussian Process (GP), our new input-dependent MFBO (iMFBO) with learnable noise models better captures the fidelity of each information source in an intuitive way. We further design a new acquisition function for iMFBO and prove that the queries selected by iMFBO have higher quality than those by naive MFBO methods, with the derived sub-linear regret bound. Experiments on both synthetic and real-world data demonstrate its superior empirical performance.

## 1 INTRODUCTION

Bayesian Optimization (BO) [Frazier, 2018] has been a powerful tool to optimize black-box functions. The term 'black-box' here indicates that we do not have access to the analytic form of either the objective function or its derivatives. We can only gain information about them by querying selected inputs to evaluate, where each evaluation can be time-consuming with prohibitive costs. Usually, BO first learns a probabilistic model, Gaussian Process (GP) [Rasmussen, 2003] for example, from available evaluations as a surrogate of the black-box objective and then iteratively selects new input(s) to query guided by some acquisition function. The acquisition function is designed to be easier to optimize compared to the original objective and achieve the desired exploration and exploitation trade-off for efficient identification of global optimum.

In Multi-fidelity Bayesian Optimization (MFBO) [Forrester et al., 2007], instead of directly evaluating expensive objective functions, we can query their less resource-demanding approximation models. Most of the existing works on MFBO consider fixed fidelity for each approximation model and optimize the underlying function within the predefined budget of cost [Kandasamy et al., 2016]. However, the fidelity of different approximation models may not be always fixed but is dependent on the input. This may arise in many adaptive reduced-order models and especially data-driven approximation models by recent machine learning (ML) methods. Typically, these approximation models tend to be more accurate in the 'data-rich' regions and less accurate in the other regions with less data.

In this work, we focus on the cases where the multi-fidelity approximations have varying fidelity for different approximation models and over the input space. We try to capture the varying fidelity by learning the input-dependent additive noise, usually ignored and considered as a hyper-parameter in many BO and MFBO methods.

Our contribution in this work is three-fold:

1. We adopted the heteroscedastic Gaussian Process to the Multi-Fidelity setup that the multi-fidelity approximations have varying fidelity over the input space as well as different input sources and proposed input-dependent MFBO (iMFBO) framework and extend it to cost-aware and bias-aware setups.

2. Based on the surrogate modeling, we proposed a new acquisition function Noise-Variant Upper Confidential Bound (NVUCB) and theoretically derived a sub-linear regret bound.

3. We further empirically compare our iMFBO with existing baselines on both synthetic and real-world examples and demonstrate its superiority.

# 2 BACKGROUND

## 2.1 SINGLE FIDELITY BAYESIAN OPTIMIZATION

For single fidelity BO (SFBO), we iteratively get noisy observations $y$ of the ground-truth objective function $f : \mathcal{X} \to \mathbb{R}$ by querying selected inputs $x$, where $y = f(x) + \delta S$, $\delta$ is noise scale, and $S \sim \mathcal{N}(0,1)$ is the standard normal white noise. BO iterates the surrogate model updating and selecting query evaluation, aiming to find the global optimum of $f(x)$ with the minimum number of queries.

In iteration $i$, we query one input $x_i$ in the input space $\mathcal{X}$, and gradually build a dataset $\mathcal{D}_t = \{(x_i, y_i)\}_{i \in \{1,2,...,t\}}$, denoting $X = [x_1, x_2, \ldots, x_t]$ and $Y = [y_1, y_2, \ldots, y_t]$. GPs are well studied probabilistic surrogate models and are commonly chosen in BO. Given $\mathcal{D}_t$, we can derive the predictive posterior assuming the GP prior for the ground-truth objective values;

$$[f(x_1), f(x_2), \ldots, f(x_t)] \sim \mathcal{N}(m, K), \forall x_i \in \mathcal{X}, \quad (1)$$

where $m$ is the mean vector (usually chosen to be 0) and $K$ is the covariance matrix with entries $K_{i,j} = k(x_i, x_j)$, where $k(\cdot, \cdot)$ is a pre-defined kernel function. The prediction at $x$ is then

$$f(x)|_{\mathcal{D}_t} \sim \mathcal{N}(\mu_t(x), \sigma_t^2(x)), \quad (2)$$

where $\mu_t(x) = K'K_t^{-1}Y$ is the posterior mean, $\sigma_t^2(x) = k(x, x) - K'K_t^{-1}K'^T$ is the posterior variance, $K_t = K + \delta^2 I$ is the covariance matrix of the observation with the observation noise $\delta$, and $K' = [k(x, x_1), k(x, x_2), \ldots, k(x, x_t)]$ [Rasmussen, 2003]. Although the observation noise $\delta$ is usually considered stationary, there have also been works on GPs considering heteroscedastic noise setups [Goldberg et al., 1997, Kersting et al., 2007, Liu et al., 2020], which inspired us to extend the input-dependent noise to the BO setup.

There are many commonly used acquisition functions in BO, such as Expected Improvement (EI) [Jones et al., 1998] and Probability of Improvement (PI) [Kushner, 1964]. Another widely studied acquisition function is the Upper Confidence Bound (UCB):

$$\alpha_t(x) = \mu_t(x) + \beta_t^{\frac{1}{2}} \sigma_t(x). \quad (3)$$

There are also many other complicated acquisition functions, especially the entropy-based ones including the Predictive Entropy Search (PES) [Hernández-Lobato et al., 2014] and Maximum-value Entropy Search (MES) [Wang and Jegelka, 2017]. These usually do not have analytic forms and require approximation or sampling methods to compute.

## 2.2 MULTI-FIDELITY BAYESIAN OPTIMIZATION

For multi-fidelity BO (MFBO), the ground-truth objective function $f$ is usually not able to be directly queried or evaluated without observation noise. Instead, we can query its different black-box approximation models, namely $f^j : \mathcal{X} \to \mathbb{R}$ at different cost $c_j$, where $j \in \mathcal{J} = \{1, 2, \ldots, J\}$ indexes the approximation functions to query. Such situations are ubiquitous in many real-world applications. Many complex systems in reality, such as the climate system, are difficult to evaluate at will. Alternative to directly measuring the actual status, we can query different surrogate models such as physics-based simulation models or data-driven ML surrogates.

Unlike in classical BO, each iteration of MFBO necessitates the selection of both an input and an approximation model for querying. Let us denote the observed data $\mathcal{D}_t^{MF} = \{(x_i, y_i^{j_i})\}_{i \in \{1,2,...,t\}}$, where $x_i$ represents the $i$th queried input and $y_i^{j_i}$ the corresponding evaluation result from the approximation model $j_i \in \mathcal{J}$. We refer $\mathcal{D}_t^{MF}$ as $\mathcal{D}_t$ in the following sections for simplicity. Note that we do not necessarily have results from all the approximation models for a given input.

## 2.3 RELATED WORK

The design of MFBO also considers two critical components as in BO: the surrogate modeling and the acquisition function derivation.

Many different probabilistic models have been proposed as surrogates in MFBO, including independent GPs for different approximation models [Lam et al., 2015], Convolved Multi-Output Gaussian Process [Zhang et al., 2017], hierarchical (co-)kriging [Shu et al., 2021, Poloczek et al., 2017], recent Bayesian Neural Networks (BNNs) [Li et al., 2020, 2021], and the Semiparametric Latent Factor Model (SLFM) [Teh et al., 2005], which is a Gaussian process-based multiple response model [Takeno et al., 2020a]. All the surrogate models previously mentioned for MFBO necessitate a pre-determined fixed fidelity. This can be achieved either by establishing correlations between approximation models as hyper-parameters, as illustrated in the works of Lam et al. [2015], Zhang et al. [2017], and Teh et al. [2005], or by utilizing the low-fidelity approximation results as inputs to the high-fidelity surrogates. The latter approach is exemplified in the research by Shu et al. [2021], Li et al. [2020], and in the later publication [Li et al., 2021].

Several MFBO acquisition functions have been derived from

their classical counterparts in BO, each tailored to different models. While there are analytical acquisition functions, such as MF-GP-UCB [Kandasamy et al., 2016], the majority tend to be entropy-based. This trend stems from the inherent nature of MFBO, where the queried evaluations originate from multiple sources. For instance, the MF-MES [Takeno et al., 2020a, Li et al., 2020, 2021] is derived from the MES, and the MF-PES [Zhang et al., 2017] is adapted from the PES used in BO.

There are also works considering different MFBO setups. For example, Song et al. [2019] proposed MF-MI-Greedy aiming at minimizing regret when querying high-fidelity evaluations is mandatory after spending a specified budget on lower-fidelity models. Kandasamy et al. [2017] considered approximation models with fixed but continuous fidelity.

Considering the observation has input-dependent noise and model it as heteroscedastic GP has previously been studied for SFBO Makarova et al. [2021], Tautvaišas and Žilinskas [2023], Griffiths et al. [2021], Kirschner and Krause [2018], sometimes termed as Risk-Averse BO, aiming at optimizing target function while restricting the risk. We extend the use of heteroscedastic GP as the surrogate model of MFBO and use the input-dependent noise to model the fidelity over different approximation models as well as the input space in this work.

Similar BO setups that allow querying different information sources have been given different names such as MFBO [Kandasamy et al., 2016] or Multi-Information-Source BO [Poloczek et al., 2017]. We use MFBO for simplicity in this paper. To contextualize our contributions within the existing literature, we present the first-ever input-dependent MFBO (iMFBO) methodology that takes into account learnable input-dependent fidelity for queried evaluations facilitated by heteroscedastic learnable noise models.

In this work, we are considering BO with different evaluation or information sources that have different "fidelity" compared to the ground truth, and the goal is to efficiently utilize the information from different sources to optimize the target function similar to the majority of MFBO work. While the term "MFBO" in the literature usually refers to BO problems with evaluation models with different costs and fixed fidelity, we want to note that this work considers a more flexible setup that is not restricted to fixed high-low fidelity but considers input-dependent fidelity to incorporate more complicated real-world scenarios.

## 3 METHOD

### 3.1 SURROGATE MODELING

Here we first present the surrogate modeling for BO with evaluations from different information sources with approx-

imation models that may have input-dependent noise. We then derive input-dependent BO methods with both single and multiple approximation models, which we respectively denote as iBO and iMFBO.

Similar to existing BO methods, we model the objective function $f$ to optimize by a GP. We would like to underscore that this study primarily focuses on scenarios where only evaluations from information sources with approximation models are accessible. In these cases, different evaluations from the corresponding approximation model may exhibit input-dependent fidelity. Consequently, the deviation from the underlying objective function $f$ can vary in accordance with the input $x$. We aim to capture input-dependent fidelity of different evaluations for more efficient BO. Most of the existing BO methods ignore the evaluation noise and model evaluations by GPs with independent additive Gaussian noise of the fixed variance $\delta^2$. Here, we explicitly model the input-dependent fidelity by additive Gaussian noise with an input-dependent variance as a random variable $\delta^2(x)$.

The predictive posterior $p(f(x)|\mathcal{D}_t)$ with GP can be derived by marginalizing out the noise variable $\delta$:

$$p(f(x)|\mathcal{D}_t) = \int_{\delta} p(f(x)|\delta, \mathcal{D}_t)p(\delta|\mathcal{D}_t)d\delta. \quad (4)$$

In this setting, the GP surrogate modeling of any input-dependent approximation model of the ground-truth objective function is $f^j(x) = f(x) + \delta_j^2(x)S$, where $j \in \mathcal{J}$ is the index of information sources, and $\delta_j(x)$ is now an input-dependent random variable and $\delta = [\delta_1, \ldots, \delta_J]$ and $S \sim \mathcal{N}(0, 1)$.

As stated before, to infer the posterior distribution of $f$, $f(x)|_{\delta, \mathcal{D}_t}$, we need to define the prior on the queried noisy evaluation(s), for which the covariance takes the following form:

$$\text{cov}[f^j(x), f^{j'}(x')] = k(x, x') + \mathbb{I}(j, j')\mathbb{I}(x, x')\delta_j^2(x), \quad (5)$$

where $\mathbb{I}(a, a')$ is the indicator function, and $\mathbb{I}(a, a') = 1$ when $a = a'$ and 0 otherwise.

At each iteration $t$, our prediction for the ground-truth objective $f(x)$ on input $x$ can be written as:

$$f_t(x)|_{\delta} := f(x)|_{\delta, \mathcal{D}_t} \sim \mathcal{N}(\mu_t(x), \sigma_t^2(x)), \quad (6)$$

where the posterior mean is $\mu_t(x) = K'\hat{K}_t^{-1}Y_t$, and $Y_t = [y_1^{j_1}, y_2^{j_2}, \ldots, y_t^{j_t}]$ denotes the previous evaluation results. The updated posterior variance becomes $\sigma_t^2(x) = k(x, x) - K'\hat{K}_t^{-1}K'^T$, and the covariance matrix of the observations becomes $\hat{K}_t = K + \Lambda(\delta_{j_1}^2(x_1), \ldots, \delta_{j_t}^2(x_t))$, where $\Lambda(\delta_{j_1}^2(x_1), \ldots, \delta_{j_t}^2(x_t))$ denotes a diagonal matrix with diagonal entries being $\delta_{j_1}^2(x_1), \ldots, \delta_{j_t}^2(x_t)$.

Here we have the form of $p(f(x)|\delta, \mathcal{D}_t)$. The other important component of the posterior (4) is $p(\delta|\mathcal{D}_t)$. Naturally, we believe that the input-dependent fidelity should have

a certain level of continuity and, in turn, can be modeled either parametrically or non-parametrically. An example illustrating the performance of these surrogate modeling can be found in Appendix B

### 3.1.1 Parametric Noise Model

One way to capture the input-dependent noise variance $\delta(x)$ is by learnable parametric models, such as linear models or MLPs, denoted by $\delta_\theta(x)$. The posterior of the objective function is then transformed to:

$$p(f(x)|\mathcal{D}_t) = \int_\theta p(f(x)|\theta, \mathcal{D}_t)p(\theta|\mathcal{D}_t)d\theta. \quad (7)$$

In this setup, we aim to learn the posterior distribution of the parameters and use the model structure to preserve the continuity of $\delta(x)$.

**Model parameter posterior $p(\theta|\mathcal{D}_t)$:** The posterior of the learnable parameters $\theta$ that model input-dependent fidelity, $p(\theta|\mathcal{D}_t)$, is another important component in (11). By Bayes' rule,

$$p(\theta|\mathcal{D}_t) \propto p(\theta)p(\mathcal{D}_t|\theta), \quad (8)$$

where $p(\theta)$ is the prior distribution of the model parameters and $p(\mathcal{D}_t|\theta)$ is the likelihood. While the prior distribution is usually selected beforehand, the likelihood takes the form:

$$p(\mathcal{D}_t|\theta) = (2\pi)^{-\frac{t}{2}}|\hat{K}_t|^{-\frac{1}{2}}\exp(-\frac{1}{2}Y_t^T(\hat{K}_t)^{-1}Y_t). \quad (9)$$

**Sampling:** Although we have the analytic forms of the prior and likelihood in our settings, Bayesian inverse to update the posterior of $\theta$ given queried evaluations usually does not have an analytic closed-form solution because of the integral in (4). One of the strategies to deal with the unnormalized distribution in (4) is by No-U-Turn Sampler (NUTS) [Hoffman et al., 2014], a variant of Hamiltonian Monte-Carlo (HMC) [Betancourt, 2017], which enables efficient sampling from unnormalized distributions.

Given the samples $\Theta = \{\theta^1, \theta^2, \ldots, \theta^M\}$ of the posterior $\theta|_{\mathcal{D}_t}$, the posterior distribution in (4) can be estimated by:

$$p(f(x)|\mathcal{D}_t) \approx \frac{1}{M}\sum_{\theta^m \in \Theta} p(f(x)|\theta^m, \mathcal{D}_t), \quad (10)$$

where $m \in 1, 2, \ldots, M$ and $M$ is the number of samples.

### 3.1.2 Non-parametric Noise Model

We also apply a non-parametric GP noise model, assuming that the available data, $\delta(x_i)$, are jointly Gaussian distributed, the posterior can be written as

$$p(f(x)|\mathcal{D}_t) = \int_{\delta_t} p(f(x)|\delta_t, \mathcal{D}_t)p(\delta_t|\mathcal{D}_t)d\delta_t, \quad (11)$$

where $p(\delta_t|\mathcal{D}_t) \propto p(\delta_t)p(\mathcal{D}_t|\delta_t)$, and $\delta_t$ denotes the random vector $[\delta_{j_1}(x_1), \ldots, \delta_{j_t}(x_t)]$.

In this non-parametric setting, we update the posterior distribution of the input-dependent variance for each input sample, and the continuity is captured by the GP prior.

GP-modeled $\sigma(x)$ will face the same intractability problem as in the parametric setup. Furthermore, $\delta(x)|_{\mathcal{D}_t}$, which plays an important role in suggesting new samples, is also intractable because of the non-Gaussian likelihood $p(\mathcal{D}_t|\delta_t)$.

Sampling methods can be time-consuming in this case because when having a batch of $M$ samples $\tilde{\delta}(x)$, computing $\delta(x)|_{\hat{\delta}_t}$ has the complexity of $\mathcal{O}(Mt^3)$ in each iteration.

In practice, we propose to apply the *Maximum a Posterior* (MAP, Murphy [2012]) point estimate $\bar{\delta}_t$ by maximizing the posterior $p(\mathcal{D}_t|\bar{\delta}_t)p(\bar{\delta}_t)$ and then use $\delta(x)|_{\bar{\delta}_t}$ to approximate $\delta(x)|_{\mathcal{D}_t}$. The corresponding estimated posterior becomes

$$p(f(x)|\mathcal{D}_t) \approx p(f(x)|\bar{\delta}_t, \mathcal{D}_t), \quad (12)$$

and $\delta(x)|_{\bar{\delta}_t}$ can be acquired by ordinary GP updates.

Comparing the posterior distributions from our setting and traditional BO settings, the main difference of our updated posterior covariance matrix $\hat{K}_t$ from the covariance $K_n$ in previous settings is that we have replaced the constant noise variance $\delta^2$ with an input-dependent noise $\delta^2(x)$. By doing this, different approximation models are dependent by modeling the covariance $k(x, x')$, instead of being mutually independent as in Kandasamy et al. [2017]. We also capture the input-dependent fidelity by observing that the correlation of the evaluations from the approximation model(s) and the ground-truth objective function is $\text{corr}(f^j(x), f(x)) = \frac{k(x,x)}{\sqrt{k(x,x)}\sqrt{k(x,x)+\delta_j(x)}} = \frac{1}{\sqrt{1+\frac{\delta_j(x)}{k(x,x)}}}$, which is again dependent on input $x$.

## 3.2 NOISE-VARIANT UCB (NVUCB)

With the previously described surrogate model updates, we now investigate the acquisition function for iBO. To achieve better sample efficiency, we first propose a new acquisition function—*Noise-Variant UCB* (NVUCB)—for single fidelity BO when the observation noise $\delta(x)$ is dependent on $x$:

$$\alpha_t^{NV}(x) = \mu_t(x) + \beta^{\frac{1}{2}}\frac{\sigma_t(x)}{\sqrt{\sigma_t^2(x) + \delta^2(x)}}\sigma_t(x). \quad (13)$$

Recall that the original UCB for BO takes the form of (3), our proposed NVUCB is basically the UCB with the standard deviation factored by $\gamma_t(x) = \frac{\sigma_t(x)}{\sqrt{\sigma_t^2(x)+\delta^2(x)}}$.

Our NVUCB acquisition function can easily be extended to multi-fidelity BO by considering multiple approximation

models, named as *Multi-Fidelity NVUCB* (MFNVUCB):

$$\alpha_t^{MFNV}(x, j) = \mu_t(x) + \beta^{\frac{1}{2}} \frac{\sigma_t(x)}{\sqrt{\sigma_t^2(x) + \delta_j^2(x)}} \sigma_t(x). \quad (14)$$

This proposed iMFBO follows the usual BO framework with our input-dependent surrogate model and MFNVUCB acquisition function, which is summarized in Algorithm 1 in Appendix A.

To explain the reason for the factorization of the deviation in (13), we go back to the idea of the original UCB (3). The first term $\mu(x)$ is designed for exploiting the surrogate estimation of potential optimal solutions and the second term encourages exploration into unknown regions in the design space.

While it is natural to consider the inferred variance $\sigma^2(x)$ as quantified model uncertainty to guide exploration, we can also be more explicit to directly consider the potential reduction of the variance after querying $x$ with noiseless observations.

Consider a potential candidate $x$, in input-dependent fidelity settings with $\mathcal{N}(0, \delta^2(x))$ observation noise, our surrogate of the ground-truth objective function at iteration $t$ is $f_t(x) \sim \mathcal{N}(\mu(x), \sigma^2(x))$, and the potential observation $f^a(x) \sim \mathcal{N}(f_t(x), \delta^2(x))$. The posterior when the observation is $y$ is then $f_t|_y \sim \mathcal{N}(\frac{\delta^2(x)\mu(x) + \sigma^2(x)y}{\delta^2(x) + \sigma^2(x)}, \frac{\delta^2(x)}{\sigma^2(x) + \delta^2(x)} \sigma^2(x))$. The variance reduction is $\text{Var}(f_t) - \text{Var}(f_t|_y) = \gamma^2(x)\sigma^2(x)$, which is exactly reflected in our factored variance term in NVUCB and MFNVUCB. In a noiseless setup, i.e. $\delta(x) = 0$, $\gamma(x) = \frac{\sigma(x)}{\sqrt{\sigma^2(x) + 0}} = 1$ so NVUCB would become original UCB. Compared to the penalty terms applied in Griffiths et al. [2021], Makarova et al. [2021], ours has similar penalty power for noisier points, but ours is derived from the information gain formulation and leads to our theoretical results. An illustration of different acquisition functions can be found in Appendix C.

Though the discussion in this section is under the unbiased evaluations assumption and equal-cost setup, we want to note that this iMFBO is capable of being extended to a bias-aware and cost-aware version and we discuss such extensions in Appendix G and F, respectively.

# 4 THEORETICAL RESULTS

As another way to illustrate the importance of the factor $\gamma(x)$, we show that it also shows up in the information gain of the ground-truth after observing noisy evaluations.

**Proposition 4.1.** *Given a set of input samples* $[x_1, x_2, \ldots, x_n]$, *the information gain of the ground-truth function* $f(x)$ *after querying approximation model* $f^a$, *with observation noise variance* $\delta^2(x)$, *getting*

$F_n^a = [f^a(x_1), f^a(x_2), \ldots, f^a(x_n)]$ *is*

$$I(f; F_n^a) = -\frac{1}{2} \sum_{i=1}^{n} \log(1 - \gamma_i^2(x_i)), \quad (15)$$

where $\gamma_i(x_i) = \frac{\sigma_t(x_i)}{\sqrt{\sigma_i^2(x_i) + \delta^2(x_i)}}$, $\sigma_i^2(x_i)$ is the predictive variance after observing $F_i^a = [f^a(x_1), f^a(x_2), \ldots, f^a(x_i)]$.

With the fact that $-\frac{1}{2} \log(1 - \gamma^2)$ is monotonically increasing with respect to $\gamma$, iBO guided by NVUCB, which encourages querying samples with larger $\gamma_t$ values at iteration $t$, is more likely resulting in more informative queries when exploring the input space. Formally, we have the following main theorem:

**Assumption 4.2.** *The ground-truth target function* $f$ *is sampled from a Gaussian Process with a kernel* $k(x, x')$.

**Theorem 4.3.** *If the latent ground-truth* $f$ *satisfies Assumption 4.2, denote* $x_v$ *as the selected candidate by the proposed NVUCB acquisition function* (13), $x_u$ *as the selected candidate by the original UCB* (3). *At least one of the following statements holds true:*

- **S1**: *The information gain of ground-truth* $f$ *after querying approximation model* $f^a$, *with observation noise variance* $\delta^2(x)$ *at* $x_v$ *can be lower bounded by that at* $x_u$, $I(f; f^a(x_v)) \geq I(f; f^a(x_u))$;

- **S2**: *The predictive mean of the selected sample* $\mu(x_v) > \mu(x_u)$.

Compared to the original UCB acquisition function, the NVUCB acquisition function would either get more informative queries (**S1**), tend to exploit the current model (**S2**), or achieve both in SFBO setup.

A sublinear regret bound can also be derived for MFNVUCB-guided iMFBO in the Multi-Fidelity setup to be $\mathcal{O}(\sqrt{\beta_T I_T^{max} T})$ under mild assumptions, following Srinivas et al. [2009]. Formally, we prove the following theorem.

**Assumption 4.4.** *The target function* $f$ *defined on* $D \subset [0, r]^d$ *is compact and convex,* $d \in N$, $r > 0$.

**Assumption 4.5.** *The kernel* $k(x, x')$ *defined in Assumption 4.2 satisfies the following high probability bound on the derivatives of GP sample paths* $f$: *There exist constants* $a, b > 0$,

$$Pr\{\sup_{x \in D} |\partial f / \partial x_k| > L\} \leq a e^{-(L/b)^2}, k = 1, \ldots, d.$$

**Assumption 4.6.** *The observation noise* $\delta_j(x)$ *for any information source* $j$ *satisfies* $\delta_{min} \leq \delta_j(x) \leq \delta_{max}$.

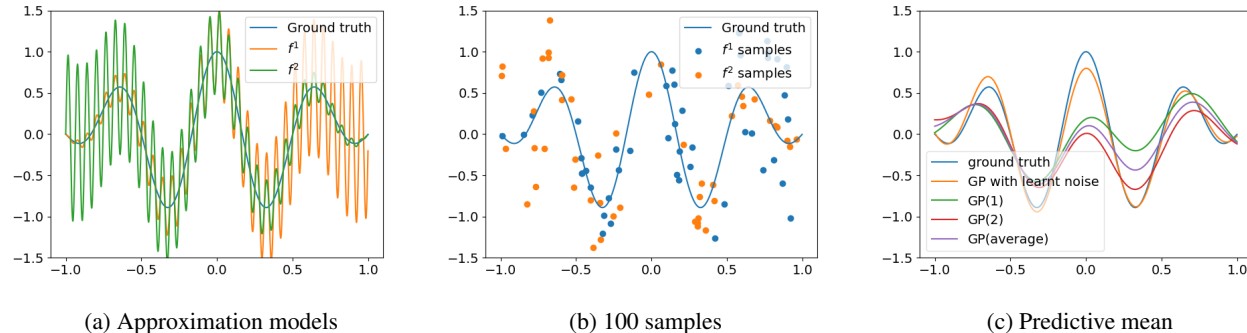

| (a) Approximation models | (b) 100 samples | (c) Predictive mean |

Figure 1: (a) The approximation model and the ground truth, though the approximation model is deterministic, we here show that the model prediction can also benefit from considering the bias as input-dependent noise. (b) Randomly drawn 100 samples, 50 from each of the approximation model. (c) The predictive mean of the surrogate models. the curve labeled with "GP with learnt noise" is the proposed GP with learnable noise, while "GP(1)" and "GP(2)" are GPs fitting approximation model $f^1$ and $f^2$ respectively. We also plotted the average value of "GP(1)" and "GP(2)" as "GP(average)".

**Theorem 4.7.** *For a constant $\epsilon \in (0,1)$, and $\beta_t = 2\log(t^2\pi^2/(3\epsilon)) + 2d\log(t^2dbr\sqrt{\log(4da/\epsilon)})$, performing MFNVUCB for a target $f$ satisfying Assumptions 4.2 4.4 4.5 with observation noise satisfying Assumption 4.6, we have*

$$Pr\{R_T \le (\sqrt{C_{\delta_{min}}}+1)\sqrt{2\delta_{max}^2\beta_T I_T^{max}T}+\frac{\pi^2}{6}\} \ge 1-\epsilon, \quad (16)$$

*where $R_T = \sum_{t=1}^T [f(x^*) - f(x_t)]$, $I_T^{max}$ is the maximum information gain at iteration $T$, and the constant $C_{\delta_{min}} > 1$ is related to $\delta_{min}$.*

The proofs for Proposition 4.1, Theorem 4.3 and Theorem 4.7 can be found in Appendix H.

We note that the learnt input-dependent noise $\delta(x)$ is considered as random variables and the acquisition functions used in the experiments are computed by taking expectation over the distribution of corresponding noise $\delta(x)$.

## 5 NUMERICAL RESULTS

We here first show that our surrogate modeling and acquisition function can capture input-dependent fidelity and approximate the ground-truth more efficiently using a toy example. We then illustrate the performance of our proposed iMFBO methods with both the toy example and well-known benchmarking optimization targets. The performance of iBO is discussed in Appendeix D. Finally, we implement NVUCB to a real-world materials discovery dataset [Zhuo et al., 2018], for which we aim to maximize the band-gap of nonmetal materials.

### 5.1 SURROGATE MODEL PERFORMANCE

We illustrate the effectiveness of capturing input-dependent fidelity by our surrogate modeling strategy with a toy ex-

ample in Figure 1, where the ground truth is $f(x) = -(x^2 - 1)\cos(3\pi x), x \in [-1, 1]$, which has two local maximums (left and right) and one global maximum (center). Two different deterministic approximation models are considered as $f^1(x) = f(x) + 0.5(x + 1)\sin(32\pi x)$, $f^2(x) = f(x) - 0.5(x - 1)\sin(32\pi x)$ that has an observation error compared to the target. We show the predictive mean of our proposed heteroscedastic noise GP (GP with inferred noise) and regular GP with constant observation noise (GP(1) and GP(2)). It can be observed that though "GP(1)" and "GP(2)" are able to correctly identify three peaks, they fail to identify the global maximum while our model can as shown in Figure 1c.

### 5.2 ACQUISITION FUNCTION ILLUSTRATION

The performance of our acquisition function is first tested with a toy example, where the ground truth is a `sin` wave over one period, $f(x) = \sin(2\pi x), x \in [0, 1]$. We consider two approximation models with the corresponding linear additive noise: $f^j(x) = f(x) + (a_j x + b_j)S$, where $a_1 = 0.5, b_1 = 0, a_2 = -0.5, b_2 = 0.5$, and $S$ denotes the standard normal distributed noise, i.e. $f^1$ is with higher fidelity when $x$ is small while $f^2$ is more precise when $x$ is large.

Compared with the MFBO proposed in Kandasamy et al. [2016], which directly applies UCB on the approximation models, we refer the acquisition function focusing only on the approximation models here as *Noise UCB* (NUCB):

$$\text{NUCB:} \quad \alpha_t^N(x) = \mu_t(x) + \beta^{\frac{1}{2}}\sqrt{\sigma_t^2(x) + \delta^2(x)}. \quad (17)$$

The acquisition function values with 20 randomly sampled data trained with the parametric linear noise model are illustrated in Figure 2. We compare the performance of NVUCB (2a) and NUCB (2b). We also fit "Separated GPs" for the two approximation models and compute the UCB acquisition function values for each of them as illustrated in

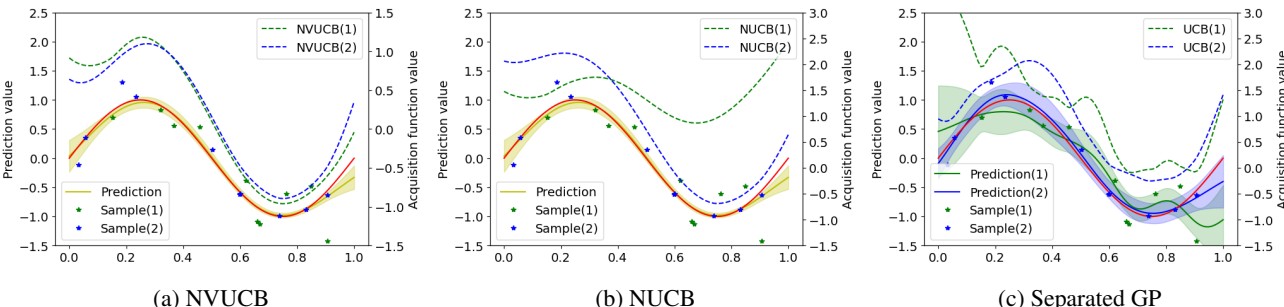

Figure 2: The corresponding acquisition functions are illustrated as dashed lines based on the learned surrogate models of the ground-truth for (a) NVUCB, (b) NUCB, and (c) Separated GP, respectively. The shaded region is 1-$\sigma$ confidence region. The number in parentheses indexes the corresponding approximation model, e.g. NVUCB(1) in (a) is by (14) with $j = 1$. The solid red line illustrates the latent ground truth $f(x)$.

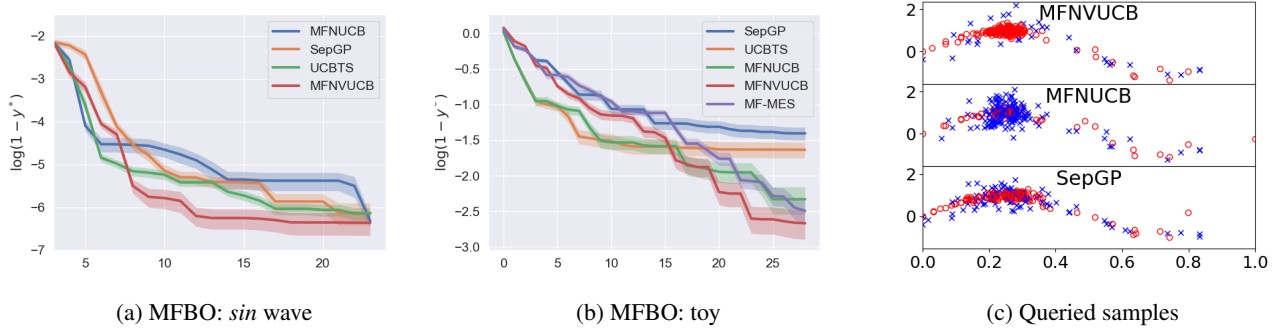

Figure 3: (a)(b) Identified maximal ground-truth values by iMFBO over 10 independent runs with random initialization. The y-axis illustrates the log-regret $\log(1 - y^*)$, where $y^*$ is the largest value queried and 1 is the optimum for both target functions, and the shaded area represents 1-$\sigma$ confidence. (c) The queried evaluations by iMFBO with different acquisition functions: the queried evaluations from $f^1$ are plotted in red and those from $f^2$ are blue.

Figure 2b. The cost of evaluating either of the two approximation models is set to 1 equally.

As the approximation model $f^1$ has higher fidelity when $x$ is small and $f^2$ is more accurate when $x$ is large, intuitively one tends to query the evaluation from the approximation model with higher fidelity. Our NVUCB acquisition, as in Figure 2a, conforms to this intuition by showing NVUCB(1) is larger when $x$ is closer to 0 and smaller when $x$ is closer to 1 compared to NVUCB(2). However, NUCB has the opposite trend and does not make use of the learnt fidelity information as in Figure 2b. Separated GPs in Figure 2c, as expected, do not properly handle the input-dependent noise and result in overestimating the model uncertainty.

We then test iMFBO with the two approximation models, with the cost of querying either of the models set to 1. We compare iMFBO with NVUCB (MFNVUCB), NUCB (MFNUCB), separated GPs (SepGP), and MF-MES [Takeno et al., 2020b]. SepGP indicates maintaining a Gaussian Process for each information source and performing BO over them. We also compare a two-step method, referred as UCBTS, in which we learn input-dependent surrogates and then choose the input by UCB and query the

approximation model for the corresponding evaluation with the lowest noise.

Figure 3a illustrates the performance assessment results of these different iMFBO methods. It is clear that our MFN-VUCB achieves the best MFBO performance and is more stable than the others. Although UCBTS shows better performance in the first few iterations, it fails to identify the global optimum with the increasing number of iterations. We have also tested the performances of the MFBO methods with the toy example in Section 5.1, which is more complicated than *sin* wave as illustrated in Figure 3b. We believe that our proposed surrogate model and acquisition function can better approximate the underlying function in this case and query better samples.

Figure 3c plots the queried evaluations from the corresponding approximation models. We can see that during iMFBO iterations, our input-dependent fidelity GPs can capture the observation noise of the two approximation models while SepGP does not show a clear preference on which approximation model to query. MFNVUCB tends to query $f^1$ near the global optimum 0.25, which has higher fidelity and is more informative. In contrast, MFNUCB does the oppo-

Table 1: Minimal ground-truth values among queried samples averaged over 10 independent runs.

|         | Hartmann 6D       | Branin            | Levy              |
|---------|-------------------|-------------------|-------------------|
| **NVUCB** | **-1.92**±0.48  | **1.26** ±1.32    | 2.009 ± 2.28      |
| NUCB    | -1.76 ± 0.41      | 7.02 ± 4.52       | 6.87 ± 7.49       |
| SepGP   | -1.79 ± 0.33      | 3.73 ± 4.18       | 2.03 ± 1.36       |
| MF-GP-UCB | -1.80 ± 0.18    | 2.66 ± 1.50       | 2.006 ± 2.33      |
| MF-MES  | –1.87 ± 0.26      | 2.19 ± 1.27       | **0.98** ± 0.80   |

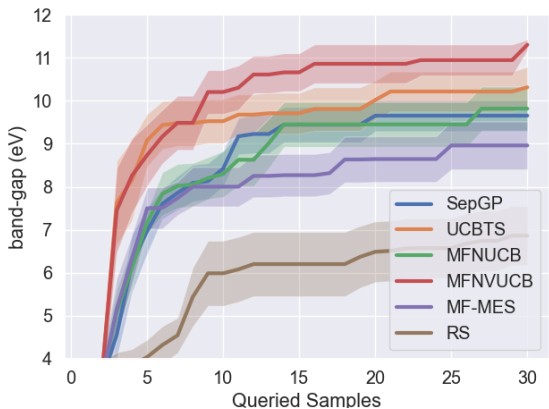

Figure 4: Maximal band-gap (eV) for queried compositions by different MFBO methods.

site, validating that our proposed surrogate modeling and NVUCB can help capture input-dependent fidelity and efficiently guide the selection of inputs and approximation models to query.

## 5.3 BENCHMARK OPTIMIZATION FUNCTIONS

In Table 1, we present the performance of our iMFBO along with MF-GP-UCB [Kandasamy et al., 2016] and MF-MES [Takeno et al., 2020b] on minimizing three benchmark objective functions: Hartmann 6D, Branin, and Levy as mean and standard deviation over 10 independent runs. In all three cases, we set two approximation models with linear additive noise, with the cost of querying either approximation model again set to 1. The detailed experimental settings can be found in Appendix I. We initialize the experiment with two randomly queried samples from each of the approximation models, i.e. four samples in total, and run the experiment for 50 iterations. For Hartmann and Branin functions, our iMFBO with the proposed surrogate modeling and NVUCB acquisition function performs the best while NUCB performs the worst, demonstrating that our surrogate model can capture the input-dependent fidelity and NVUCB utilizes it efficiently. Our iMFBO with NVUCB also performs comparably well to MF-GP-UCB when optimizing the Levy benchmark function.

## 5.4 MATERIALS DISCOVERY PROBLEM

We now test our iMFBO on a real-world materials dataset, for which we must find the material with the largest band-

gap.

### 5.4.1 Data Collection

We consider the experimental dataset used and reported by Zhuo et al. [2018] as the ground truth $f$. This data set consists of 3,896 experimentally-characterized band-gap measurements from 2,458 unique inorganic compounds. These experimental band-gaps were obtained using a number of experimental techniques, including diffuse reflectance, resistivity measurements, surface photovoltage, photoconduction, and UV–vis measurements.

Density Functional Theory (DFT) calculations are often used to predict a variety of material properties, including the band-gap [Jain et al., 2013]. These predictions can vary based on the structural and other configurations of the material. We take the band-gap characterized by the smallest energy per atom as reported in the open-access Material Project (MP) dataset [Jain et al., 2013], which is considered a suitable approximation [Ward et al., 2018] based on the principle that the compound with the lowest energy per atom corresponds to the predicted ground state for that particular chemical system. By querying the MP with the compositions present in the ground-truth dataset, we successfully identified DFT band-gap calculations for 1,439 out of the 2,458 compositions. DFT band-gap values were not available in the MP for the remaining 1,019 compositions, which we use to train another data-driven surrogate based on the experimental measurements.

### 5.4.2 Preprocessing

In dealing with duplicated compositions from the dataset provided by Zhuo et al. [2018], we opted to retain only the first reported entry, due to the absence of corresponding experimental conditions and energy per atom values within the dataset.

We consider querying three approximation models: (1) the DFT-calculated band-gap $f_1$ in MP, (2) the band-gap predicted by a pre-trained MLP $f_2$, and (3) the band-gap predicted by a pre-trained linear model $f_3$. The MLP and linear models are trained based on the 1,019 compositions without DFT band-gap values recorded in MP. The objective of iMFBO is to identify the material with the largest band-gap among the other 1,439 compositions. The experiments are initiated with one random sample from each approximation model. We evaluate performance over 20 independent runs with a budget ($B$) of 30, as illustrated in Figure 4. It is important to note that all queries in this experiment can only yield approximations (either from DFT or MLP evaluations), and performance is assessed based on the experimental band-gaps reported in Zhuo et al. [2018]. The details about the pre-trained MLP are provided in Appendix J. We use the input of the MLP's final prediction layer as a feature extractor, distilling the original features down to two. All MFBO methods are performed in this extracted 2-dimensional space.

As shown in Figure 4, our MFNVUCB outperforms other models as anticipated. Interestingly, our UCBTS consistently surpasses SepGP, and we observe that MFNUCB yields superior final performance to SepGP within the budget. We also reported the performance of the random selection policy (RS) demonstrating the effectiveness of the BO methods. This outcome likely stems from our input-dependent fidelity surrogate modeling, which utilizes queried evaluations more efficiently in conjunction with the corresponding acquisition functions.

## 6 CONCLUSION

In this study, we have introduced iMFBO, an innovative Multi-Fidelity Bayesian Optimization method. This approach models the input-dependent fidelity of each approximation model, formulates a novel acquisition function, NVUCB, and is capable of being extended to cost-aware and bias-aware setups. Our framework integrates the learned input-dependent fidelity to more effectively guide the adaptive query evaluation of corresponding approximation models in each iteration. Our method is particularly suited to many scientific problems, such as materials discovery, where multiple information sources are available, each providing insights into the ground truth at varying levels of fidelity. We evaluated our proposed iMFBO on both synthetic and real-world datasets, demonstrating its proficiency in capturing the input-dependent fidelity of multiple approximation models and its efficiency in optimizing the underlying ground-truth objective function based on approximation evaluations. Therefore, this work underscores the potential of iMFBO in effectively addressing multi-fidelity optimization problems, particularly in complex scientific fields where diverse sources of information must be synthesized.

## ACKNOWLEDGEMENTS

This work was supported in part by the U.S. National Science Foundation (NSF) grants CCF-1553281, DMREF-2119103, SHF-2215573, and IIS-2212419; and by the U.S. Department of Engergy (DOE) Office of Science, Advanced Scientific Computing Research (ASCR) M2DT Mathematical Multifaceted Integrated Capability Center (MMICC) under Award B&R# KJ0401010/FWP# CC130, program manager W. Spotz. Portions of this research were conducted with the advanced computing resources provided by Texas A&M High Performance Research Computing.

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

# Appendix
# (Supplementary Material)

**Mingzhou Fan**[1]     **Byung-Jun Yoon**[1,2]     **Edward Dougherty**[1]     **Nathan Urban**[2]

**Francis Alexander**[3]     **Raymundo Arróyave**[4,5]     **Xiaoning Qian**[1,2,6]

[1]Department of Electrical & Computer Engineering, Texas A&M University, College Station, Texas, USA,
[2]Computational Science Initiative, Brookhaven National Laboratory, Upton, New York, USA,
[3]Argonne National Laboratory, Lemont, Illinois, USA,
[4]Department of Materials Science & Engineering, Texas A&M University, College Station, Texas, USA
[5]Department of Mechanical Engineering, Texas A&M University, College Station, Texas, USA
[6]Department of Computer Science & Engineering, Texas A&M University, College Station, Texas, USA

## A   IMFBO ALGORITHM

We present our proposed iMFBO pseudo-code as follows:

---
**Algorithm 1** iMFBO
---
    **Initialize** Initial dataset $\mathcal{D}_t = \mathcal{D}_0$, budget $B$, time step $t = 1$;
    **repeat**
        Fit the surrogate model of the latent ground truth and input-dependent fidelity to the current dataset $\mathcal{D}_t$;
        Find $(x_t, j_t)$ pair that maximize the equation (14);
        Query the $j$-th approximation model on sample $x_t$;
        Update the dataset $\mathcal{D}_{t+1} = \mathcal{D}_t \cup (x_t, y_t^{j_t})$
        Update budget $B = B - 1$;
        Update time step $t = t + 1$;
    **until** $B \leq 0$
---

## B   PERFORMANCE OF THE NOISE SURROGATE

We illustrate the effectiveness of capturing input-dependent fidelity by our surrogate modeling strategy with a toy example, where the ground truth is a `sin` wave over one period, $f(x) = \sin(2\pi x), x \in [0, 1]$. We consider two approximation surrogate models with the corresponding linear additive noise: $f^j(x) = f(x) + (a_j x + b_j)S$, where $a_1 = 0.5$, $b_1 = 0$, $a_2 = -0.5$, $b_2 = 0.5$, and $S$ is the standard normal distributed noise, i.e. $f^1$ is with higher fidelity when $x$ is small while $f^2$ is more precise when $x$ is large.

In parametric surrogate modeling (Sec. 3.1.1), we can take a linear noise model. The priors of the bias and weights are set to be standard normal distributions for both approximation models. The posterior of $\delta(x)$ trained with 20 and 500 random evaluation samples are illustrated in Figures 5a and 5b. In the non-parametric setting, as illustrated in Figure 5c, the prior of the noise model is set to be a Gaussian Process with zero mean and Radial Basis Function (RBF) covariance kernel [Seeger, 2004].

When acquiring more evaluations from approximation models, the uncertainty of the noise model reduces from Figure 5a to Figure 5b, indicating that our model can reliably capture the model uncertainty. When we have a large queried evaluation set, the learnt noise fits the ground truth well in Figure 5b. More importantly, even with a relatively limited number of queried approximation evaluations, the input-dependent noise trend can still be reliably learned as in Figure 5a. The trend of noise is also learnt reasonably well with the non-parametric surrogate modeling as shown in Figure 5c.

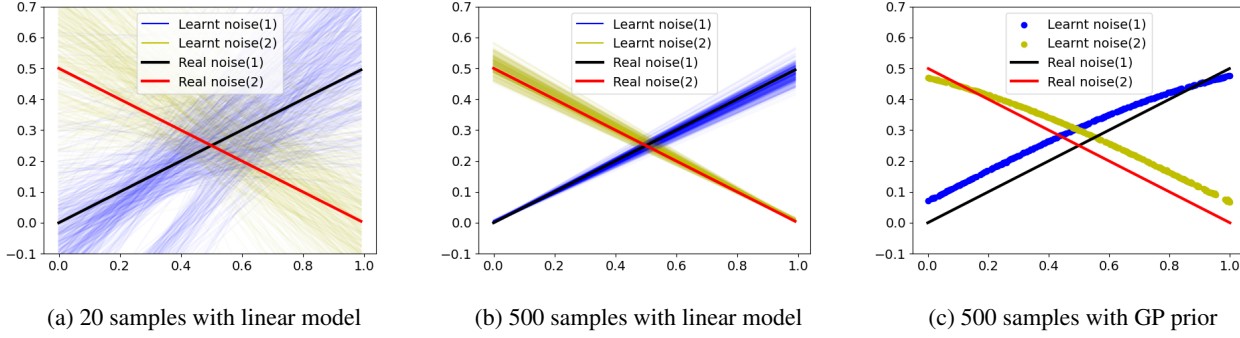

| (a) 20 samples with linear model | (b) 500 samples with linear model | (c) 500 samples with GP prior |

Figure 5: (a) The learnt noise with 20 random samples with the linear model. Each line is a sample generated by the updated parameter posterior. (b) The learnt linear noise model with 500 random samples. (c) The MAP estimation of the input-dependent noise over 500 random samples with the Gaussian Process prior. The number in parentheses indexes the approximation model, e.g., "Learnt noise(1)" in (a) is the learnt observation noise of the approximation model $f^1$.

## C   AN EXAMPLE TO ILLUSTRATE THE EFFECTIVENESS OF NVUCB

We here use an example to illustrate the effectiveness of our proposed acquisition function, NVUCB, in Figure 6. The controlling parameter $\beta$ is set to 1 for all of the three acquisition functions: UCB, NUCB, and NVUCB. Consider we have three candidate samples at $x_1, x_2, x_3$ to query. Comparing $x_2$ and $x_3$, with the same prediction mean 0 and prediction variance 12.5 by the approximation model, NUCB will face a tie. Both UCB and NVUCB break the tie by selecting more informative sample $x_3$. However, $x_1$, with predictive mean 0.5 and less observation noise variance $0.1^2$, would have the same UCB value as the one at $x_2$. NVUCB would select candidate $x_1$ which not only has higher prediction mean but also is more informative with $\gamma(x_1) = \frac{2}{\sqrt{2^2+0.1^2}} > \frac{2.5}{\sqrt{12.5}} = \gamma(x_3)$.

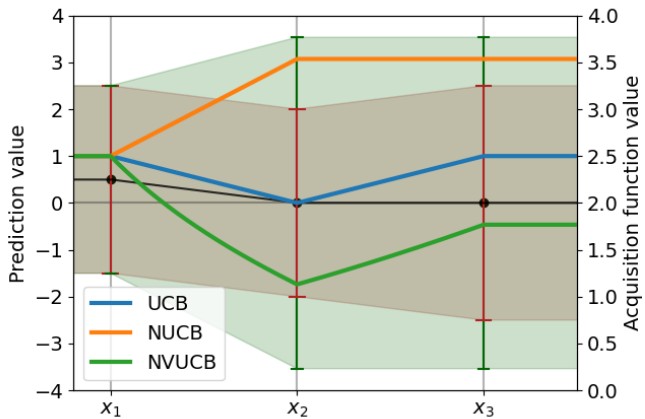

Figure 6: An illustration of UCB, NUCB and NVUCB acquisition functions. The predictive mean is plotted in solid black, The red shaded region represents the $1 - \sigma$ confidence of the ground-truth model and the green shaded region represents the $1 - \sigma$ confidence of the approximation model. NVUCB guides towards the best selection.

## D   ACQUISITION FUNCTION PERFORMANCE ON SFBO

We further test the performance of our proposed iBO using the corresponding single approximation model over 10 independent runs, as illustrated in Figures 7a and 7b. In each run, we randomly select four samples in the design space of interval $x \in [0, 1]$ for the initial dataset with two queried from $f^1$ and $f^2$ respectively. We compare iBO with NVUCB and NUCB to BO with UCB based on ordinary GP surrogates with constant observation noise. For an ablation study, we further

test the performance of BO with UCB based on our input-dependent GP surrogates, referred to as iUCB.

By comparing Figures 7a and 7b, we can observe that all methods perform better using $f^1$ evaluations than using $f^2$. This is because $f^1$ has higher fidelity in the relatively well-performing region (near the optimum $x = 0.25$) and can be more informative for BO to find the global maximum. The better performance of iUCB compared to UCB demonstrates that our input-dependent fidelity GP models the ground-truth objective better than constant noise GP. With only $f^2$ evaluations, iBO with NVUCB performs slightly worse than iUCB since NVUCB would encourage to query the inputs with smaller observation noise of $f^2$ evaluations, where the ground-truth objective values are smaller, hence hindering the maximization task.

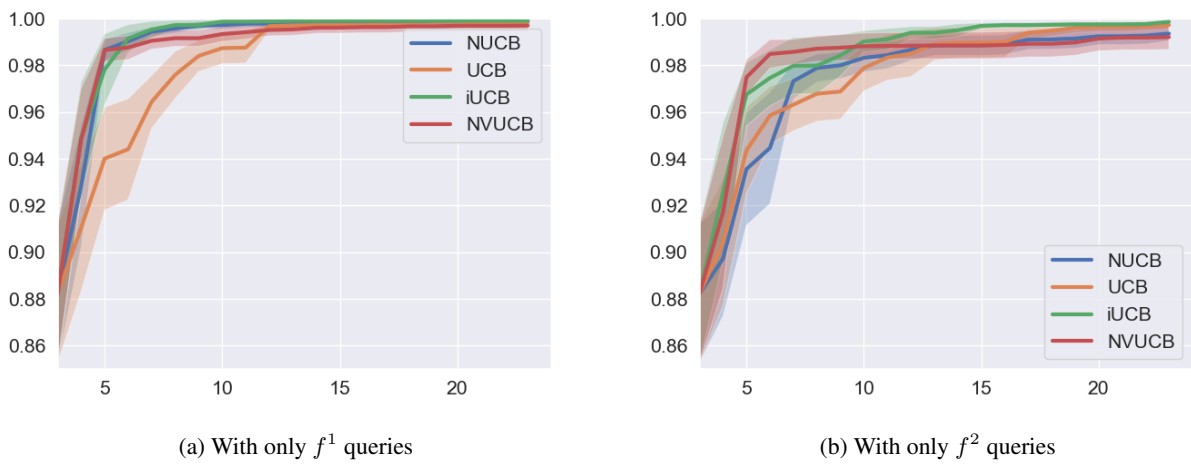

(a) With only $f^1$ queries

(b) With only $f^2$ queries

Figure 7: Identified maximal ground-truth values by iBO with different surrogates and acquisition functions over 10 independent runs with random initialization. The error bar represents the $1 - \sigma$ confidence interval.

# E  EXTENDED EXPERIMENTAL RESULTS FOR THE MATERIALS DISCOVERY PROBLEM

To better demonstrate our method, we extend the previous experiments in Section 5.4 by only considering querying the first two approximation models: (1) the DFT-calculated band-gap $f_1$ in MP, and (2) the band-gap predicted by a pre-trained MLP $f_2$. Similar to the previous experiment, the MLP alse is trained based on the 1,019 compositions without DFT band-gap values recorded in MP. The objective of iMFBO is to identify the material with the largest band-gap among the other 1,439 compositions. The experiments are initiated with one random sample from each approximation model. We evaluate

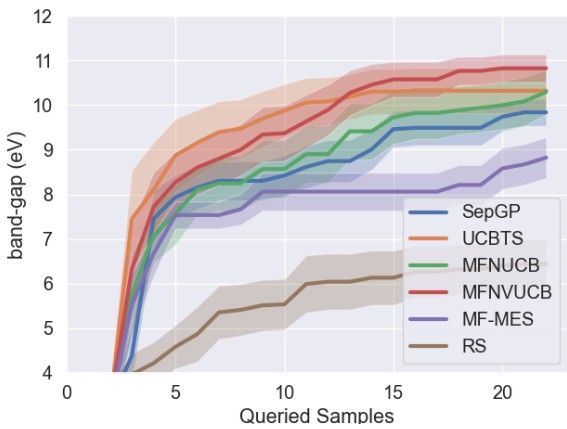

Figure 8: Maximal band-gap (eV) for queried compositions by different MFBO methods.

performance over 20 independent runs with a budget ($B$) of 20, as illustrated in Figure 8.

As shown in Figure 8, similar to the previous results, our MFNVUCB outperforms other models as anticipated. The performance of other methods also performs similarly as discussed before.

## F   IMFBO CONSIDERING EVALUATION COST

To incorporate the fact that the information resources of different approximate evaluation models usually have different evaluation costs, we can further modify the acquisition function to

$$\alpha_t^{MFNVC}(x, j) = \mu_t(x) + \frac{1}{c_j}\beta^{\frac{1}{2}}\frac{\sigma_t(x)}{\sqrt{\sigma_t^2(x) + \delta_j^2(x)}}\sigma_t(x), \tag{18}$$

where $c_k$ is the cost to evaluate each of the approximation models. The reason that we put the cost on the standard deviation term is that our surrogates of the approximation models are only different on the input-dependent noise $\delta(x)$, which only appears in the variance reduction term in the acquisition function.

We present our proposed iMFBO considering such costs as follows:

---

**Algorithm 2** iMFBO with cost

---

    **Initialize** Initial dataset $\mathcal{D}_t = \mathcal{D}_0$, budget $B$, time step $t = 1$;
    **repeat**
        Fit the surrogate model of the latent ground-truth and input-dependent fidelity to the current dataset $\mathcal{D}_t$;
        Find $(x_t, j_t)$ pair that maximize the equation (18);
        Query the $j$-th approximation model on sample $x_t$;
        Update the dataset $\mathcal{D}_{t+1} = \mathcal{D}_t \cup (x_t, y_t^{j_t})$
        Update budget $B = B - c_{j_t}$;
        Update time step $t = t + 1$;
    **until** $B \leq 0$

---

To numerically test the performance of the iMFBO implementation considering evaluation cost, we test it with the benchmark in the materials discovery problem (Section 5.4).

To reflect the reality that DFT computations are typically more resource-intensive than querying machine learning models, we set the cost for querying DFT computations and MLP to be 5 and 1, respectively.

As shown in Figure 9, the performance trends of different competing models are similar as reported in Section 5.4 when the querying costs are taken into consideration, with our MFNVUCB-based iMFBO outperforming other models as anticipated.

## G   IMFBO CONSIDERING EVALUATION BIAS

Though we consider unbiased evaluations in the main text, it is capable of extending the framework to multi-fidelity evaluations with bias. By considering the surrogate model for each evaluation model or information source $f_i(x)$ as the addition of the ground truth $f(x)$ and a separate bias $g_i(x)$ modeled by a separate model, i.e. $f_i(x) = f(x) + g_i(x) + \delta_i(x)S$. The posterior of the bias $g_i$ can also be inferred based on Bayes' rule similar to the inference of $\delta_i$ in the main text.

Here we use an illustrative experiment similar to the one in Section 5.2 to demonstrate the performance of iMFBO considering bias. The target function is set to be a sine wave $f(x) = \sin(2\pi x)$, and the evaluation models as multiple information sources are biased such that $f_1(x) = f(x) + 0.5 + 0.5xS$, $f_2(x) = f(x) - 0.5 + (-0.5x + 0.5)S$. Similar to the previous experiments, we use a GP to estimate the ground truth, the input-dependent noise for each information source $\delta_i(x)$ is estimated by a linear model, and the bias is modeled as a constant in this case, and estimated by MAP.

We illustrate the performance in Figure 10. It can be observed that our method (MFNVUCB) constantly outperforms other methods, indicating that our framework is capable to be extend to bias-aware versions.

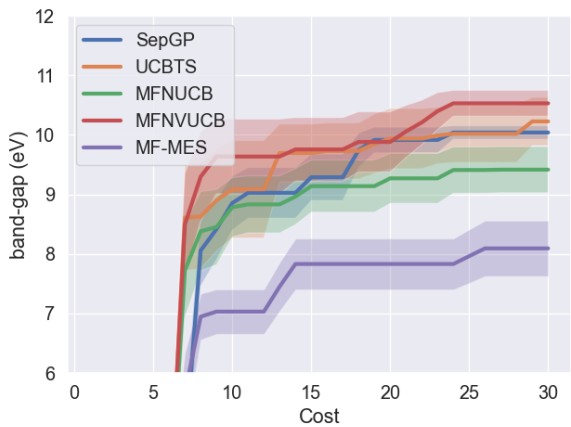

Figure 9: Maximal band-gap (eV) for queried compositions by different MFBO methods.

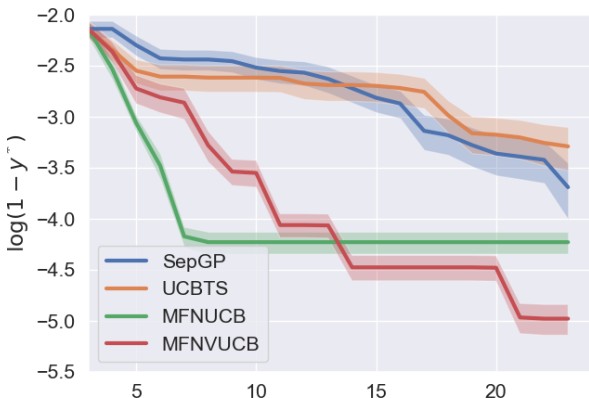

Figure 10: Log regret for biased evaluations.

# H   PROOFS

**Proposition H.1.** *Given a set of input samples* $[x_1, x_2, \ldots, x_n]$*, the information gain of the ground-truth function* $f(x)$ *after querying approximation model* $f^a$*, with observation noise variance* $\delta^2(x)$*, getting* $F_n^a = [f^a(x_1), f^a(x_2), \ldots, f^a(x_n)]$ *is*

$$I(f; F_n^a) = -\frac{1}{2} \sum_{i=1}^{n} \log(1 - \gamma_i^2(x_i)), \tag{19}$$

*Proof.* By the definition of entropy, we can derive:

$$I(f; F_n^a) = H(f) - H(f|F_n^a) = H(F_n^a) - H(F_n^a|f). \tag{20}$$

By the independent observation noise assumption,

$$H(F_n^a|f) = H(F_n^a|F_n), \tag{21}$$

where $F_n = [f(x_1), f(x_2), \ldots, f(x_n)]$; and the conditional entropy term $H(F_n^a|F_n) = \frac{1}{2} \sum_{i=1}^{n} \log(2\pi e \delta(x_i))$, again by the independent noise assumption.

The entropy term $H(F_n^a)$ can be recursively calculated as

$$\begin{aligned}
H(F_n^a) &= H(F_{n-1}^a) + H(f^a(x_n)|F_{n-1}^a) \\
&= H(F_{n-1}^a) + \frac{1}{2}\log[2\pi e(\sigma_n^2(x_n) + \delta^2(x_n))] \\
&= \frac{1}{2}\sum_{i=1}^{n}\log[2\pi e(\sigma_i^2(x_i) + \delta^2(x_i))].
\end{aligned}$$

Combining these two terms, we get the information gain with an analytic form as in (19). $\qquad\square$

**Lemma H.2.** *With the same setup as Theorem 4.3, the information gain can be lower bounded by*

$$I(f; f^a(x_v)) \geq I(f; f^a(x_u)) + \frac{\gamma(x_u)[\sigma(x_v) - \sigma(x_u)]}{[1 + \gamma(x_u)]\sigma(x_v)}, \tag{22}$$

*where $\gamma(x) = \frac{\sigma(x)}{\sqrt{\sigma^2(x) + \delta^2(x)}}$.*

*Proof.* By the assumptions that $x_v$ maximizes (13),

$$\mu(x_v) + \beta^{1/2}[\gamma(x_v)\sigma(x_v) - \gamma(x_u)\sigma(x_u)] \geq \mu(x_u), \tag{23}$$

and $x_u$ maximizes (3),

$$\mu(x_u) \geq \mu(x_v) + \beta^{1/2}\sigma(x_v) - \beta^{1/2}\sigma(x_u). \tag{24}$$

Combining (23) and (24) gives us:

$$\gamma(x_v)\sigma(x_v) - \gamma(x_u)\sigma(x_u) \geq \sigma(x_v) - \sigma(x_u), \tag{25}$$

which can be rewritten as

$$\gamma(x_v) \geq \gamma(x_u) + \frac{\sigma(x_v) - \sigma(x_u)}{\sigma(x_v)}[1 - \gamma(x_u)]. \tag{26}$$

With Proposition 4.1, for any $x$,

$$I(f; f^a(x)) = H(f) - H(f|f^a(x)) = -\frac{1}{2}\log[1 - \gamma^2(x)]. \tag{27}$$

By Jensen's inequality and the fact that the function $-\frac{1}{2}\log(1 - \gamma^2)$ is convex with respect to $\gamma$,

$$I(f; f^a(x_u)) \geq I(f; f^a(x_v)) + G(x_u, x_v), \tag{28}$$

where $G(x_u, x_v) = \frac{\gamma(x_u)}{1 - \gamma^2(x_u)}\frac{\sigma(x_v) - \sigma(x_u)}{\sigma(x_v)}[1 - \gamma(x_u)] = \frac{\gamma(x_u)[\sigma(x_v) - \sigma(x_u)]}{[1 + \gamma(x_u)]\sigma(x_v)}$. $\qquad\square$

**Lemma H.3.** *With the same setup as in Theorem 4.3, when $\sigma(x_v) \leq \sigma(x_u)$, the information gain can also be lower bounded by:*

$$I(f; f^a(x_v)) \geq I(f; f^a(x_u)) + \frac{\gamma(x_u)[\mu(x_u) - \mu(x_v)]}{[1 - \gamma^2(x_u)]\beta^{\frac{1}{2}}\sigma(x_v)}. \tag{29}$$

*Proof.* By the assumptions that $x_v$ maximizes (13),

$$\begin{aligned}
\mu(x_v) &\geq \mu(x_u) + \beta^{1/2}[\gamma(x_u)\sigma(x_u) - \gamma(x_v)\sigma(x_v)] \\
&= \mu(x_u) + \beta^{1/2}[\gamma(x_u)\sigma(x_u) - \gamma(x_u)\sigma(x_v)] \\
&\quad + \beta^{1/2}[\gamma(x_u)\sigma(x_v) - \gamma(x_v)\sigma(x_v)] \\
&\geq \mu(x_u) + \beta^{1/2}[\gamma(x_u)\sigma(x_v) - \gamma(x_v)\sigma(x_v)] \\
&= \mu(x_u) + \beta^{1/2}[\gamma(x_u) - \gamma(x_v)]\sigma(x_v).
\end{aligned} \tag{30}$$

The last step can be derived by the assumption that $\sigma(x_v) \leq \sigma(x_u)$. Therefore we now have:

$$\gamma(x_v) \geq \gamma(x_u) + \frac{\mu(x_u) - \mu(x_v)}{\beta^{\frac{1}{2}}\sigma(x_v)}. \tag{31}$$

Similar as Lemma H.2, by Jensen's inequality,

$$I(f; f^a(x_v)) \geq I(f; f^a(x_u)) + \frac{\gamma(x_u)}{1 - \gamma^2(x_u)}\frac{\mu(x_u) - \mu(x_v)}{\beta^{\frac{1}{2}}\sigma(x_v)}. \tag{32}$$

$\square$

**Theorem H.4.** *If the latent ground-truth $f$ satisfies Assumption 4.2, denote $x_v$ as the selected candidate by the proposed NVUCB acquisition function (13), $x_u$ as the selected candidate by the original UCB (3). At least one of the following statements holds true:*

- ***S1**: The information gain of ground-truth $f$ after querying approximation model $f^a$, with observation noise variance $\delta^2(x)$ at $x_v$ can be lower bounded by that at $x_u$, $I(f; f^a(x_v)) \geq I(f; f^a(x_u))$;*
- ***S2**: The predictive mean of the selected sample $\mu(x_v) > \mu(x_u)$.*

Compared to the original UCB acquisition function, the NVUCB acquisition function would either get more informative queries (**S1**), tend to exploit the current model (**S2**), or achieve both.

*Proof.* We only need to prove that **S1** holds when **S2** does not.

When **S2** does not hold, i.e. $\mu(x_v) \leq \mu(x_u)$:

1. If $\sigma(x_v) \geq \sigma(x_u)$, $I(f; f^a(x_v)) \geq I(f; f^a(x_u))$ by Lemma H.2;
2. If $\sigma(x_v) < \sigma(x_u)$, $I(f; f^a(x_v)) \geq I(f; f^a(x_u))$ by Lemma H.3.

Therefore, we can conclude that $I(f; f^a(x_v)) \geq I(f; f^a(x_u))$ if $\mu(x_v) \leq \mu(x_u)$, which proves that at least one of **S1** and **S2** is true. $\square$

We would also like to first theoretically compare the performance of NUCB with our NVUCB in the single fidelity scenario.

**Theorem H.5.** *With the same setup as Theorem 4.3, denote $x_v$ as the selected candidate by our proposed NVUCB acquisition function (13), $x_n$ as the selected candidate by NUCB (17). At least one of the following statements holds true:*

- ***T1**: The information gain $I(f; f^a(x_v)) \geq I(f; f^a(x_n))$;*
- ***T2**: The predictive mean of the selected sample $\mu(x_v) > \mu(x_n)$.*

*Proof.* As $x_v$ maximizes (13),

$$\mu(x_v) + \beta^{1/2}[\gamma(x_v)\sigma(x_v) - \gamma(x_n)\sigma(x_n)] \geq \mu(x_n), \tag{33}$$

and $x_n$ maximizes (17),

$$\begin{aligned}\mu(x_n) \geq &\mu(x_v) + \beta^{1/2}\sqrt{\sigma^2(x_v) + \delta^2(x_v)} \\ &- \beta^{1/2}\sqrt{\sigma^2(x_n) + \delta^2(x_n)}.\end{aligned} \tag{34}$$

The standard deviation term $\sqrt{\sigma^2(x) + \delta^2(x)}$ can also be written as $\frac{\sigma(x)}{\gamma(x)}$, combining (33) and (34), we have

$$\begin{aligned}&\gamma(x_v)\sigma(x_v) - \gamma(x_n)\sigma(x_n) \\ \geq &\sqrt{\sigma^2(x_v) + \delta^2(x_v)} - \sqrt{\sigma^2(x_n) + \delta^2(x_n)} \\ = &\frac{\sigma(x_v)}{\gamma(x_v)} - \frac{\sigma(x_n)}{\gamma(x_n)}.\end{aligned} \tag{35}$$

Rewriting (35), we have

$$[\gamma(x_v) - \frac{1}{\gamma(x_v)}]\sigma(x_v) \geq [\gamma(x_n) - \frac{1}{\gamma(x_n)}]\sigma(x_n). \tag{36}$$

Similar as in the proof of Theorem 4.3, we only need to prove that **T1** holds when **T2** does not.

When **T2** does not hold, i.e. $\mu(x_v) \leq \mu(x_n)$:

1. If $\sigma(x_v) \geq \sigma(x_n)$, recall that $0 < \gamma(x) < 1$, so $\gamma(x_v) - \frac{1}{\gamma(x_v)} < 0$, we have $\gamma(x_v) - \frac{1}{\gamma(x_v)} \geq \gamma(x_n) - \frac{1}{\gamma(x_n)}$ from (36). We can further have $\gamma(x_v) \geq \gamma(x_n)$ because of the monotonicity of function $\gamma - \frac{1}{\gamma}$.

2. If $\sigma(x_v) < \sigma(x_n)$,

$$
\begin{aligned}
&\gamma(x_v) - \gamma(x_u) \\
=&\sigma^{-1}(x_n)[\gamma(x_v)\sigma(x_n) - \gamma(x_u)\sigma(x_n)] \\
>&\sigma^{-1}(x_n)\gamma(x_v)\sigma(x_v) - \gamma(x_u)\sigma(x_n) \\
\geq&\sigma^{-1}(x_n)\beta^{-\frac{1}{2}}[\mu(x_n) - \mu(x_v)] \\
\geq&0.
\end{aligned} \tag{37}
$$

We can then conclude that when $\mu(x_v) \leq \mu(x_n)$, we have $\gamma(x_v) \geq \gamma(x_u)$, and can further get $I(f; f^a(x_v)) \geq I(f; f^a(x_n))$ because of the monotonicity of (19).

Therefore we have proven the theorem. $\qquad \square$

Note that Theorem 4.3 can also be proven with a similar strategy as in the proof of Theorem H.5 without using Lemma H.3 and Lemma H.3. Those two lemmas, however, allow us to better understand the acquisition functions from the information-theoretic point of view.

**Theorem H.6.** *For a constant $\epsilon \in (0,1)$, and $\beta_t = 2\log(t^2\pi^2/(3\epsilon)) + 2d\log(t^2 dbr\sqrt{\log(4da/\epsilon)})$, performing MFNVUCB for a target $f$ satisfying Assumptions 4.2 4.4 4.5 with observation noise satisfying Assumption 4.6, we have*

$$Pr\{R_T \leq (\sqrt{C_{\delta_{min}}} + 1)\sqrt{2\delta_{max}^2\beta_T I_T^{max}T} + \frac{\pi^2}{6}\} \geq 1 - \epsilon, \tag{38}$$

*where $R_T = \sum_{t=1}^{T}[f(x^*) - f(x_t)]$, $I_T^{max}$ is the maximum information gain at iteration $T$, and the constant $C_{\delta_{min}} > 1$ is related to $\delta_{min}$.*

*Proof.* Based on Lemma 5.7 in Srinivas et al. [2009], let $[x^*]_t$ be the closest point in $D_t$ to $x^*$, where $D_t$ is the discredited subset of $D$,

$$|f(x^*) - \mu_t([x^*]_t)| \leq \beta_t^{\frac{1}{2}}\sigma_{t-1}([x^*]_t) + \frac{1}{t^2} \tag{39}$$

holds with probability greater or equal to $1 - \delta$. Note that our notation of time index is different from Srinivas et al. [2009].

By definition of $x_t$ and $j_t$,

$$\mu_t(x_t) + \beta^{\frac{1}{2}}\gamma_t^{j_t}(x_t)\sigma_t(x_t) \geq \mu_t([x^*]_t) + \beta^{\frac{1}{2}}\gamma_t^{j_t^*}([x^*]_t)\sigma_t([x^*]_t), \tag{40}$$

The index on the RHS $j_t^*$ can be chosen arbitrarily so we omit it in the following derivation. So regret in one iteration:

$$
\begin{aligned}
r_t &= f(x^*) - f(x_t) \\
&\leq \mu_t([x^*]_t) + \beta_t^{\frac{1}{2}} \sigma_t([x^*]_t) + 1/t^2 - f(x_t) \quad \text{(by Equation 39)} \\
&= \mu_t([x^*]_t) + \beta_t^{\frac{1}{2}} \gamma_t([x^*]_t)\sigma_t([x^*]_t) + \beta_t^{\frac{1}{2}}(1 - \gamma_t([x^*]_t))\sigma_t([x^*]_t) + 1/t^2 - f(x_t) \\
&\leq \mu_t(x_t) + \beta_t^{\frac{1}{2}} \gamma_t(x_t)\sigma_t(x_t) + \beta_t^{\frac{1}{2}}(1 - \gamma_t([x^*]_t))\sigma_t([x^*]_t) + 1/t^2 - f(x_t) \quad \text{(by Equation 40)} \\
&\leq \beta_t^{\frac{1}{2}} \sigma_t(x_t) + \beta_t^{\frac{1}{2}} \gamma_t(x_t)\sigma_t(x_t) + \beta_t^{\frac{1}{2}}(1 - \gamma_t([x^*]_t))\sigma_t([x^*]_t) + 1/t^2 \quad \text{(by Lemma 5.1 in Srinivas et al. [2009])} \\
&= \beta_t^{\frac{1}{2}}(1 + \gamma_t(x_t))\delta_{max}\frac{\sigma_t(x_t)}{\delta_{max}} + \beta_t^{\frac{1}{2}}(1 - \gamma_t([x^*]_t))\delta_{max}\frac{\sigma_t([x^*]_t)}{\delta_{max}} + 1/t^2 \\
&\leq \beta_t^{\frac{1}{2}}(1 + \gamma_t(x_t))\delta_{max}\frac{\sigma_t(x_t)}{\delta(x_t)} + \beta_t^{\frac{1}{2}}(1 - \gamma_t([x^*]_t))\delta_{max}\frac{\sigma_t([x^*]_t)}{\delta_t([x^*]_t)} + 1/t^2 \\
&= \delta_{max}\beta_t^{\frac{1}{2}}(1 + \gamma_t(x_t))\frac{\gamma_t(x_t)}{\sqrt{1 - \gamma_t^2(x_t)}} + \delta_{max}\beta_t^{\frac{1}{2}}(1 - \gamma_t([x^*]_t))\frac{\gamma_t([x^*]_t)}{\sqrt{1 - \gamma_t^2([x^*]_t)}} + 1/t^2 \\
&= \delta_{max}\beta_t^{\frac{1}{2}}\gamma_t(x_t)\sqrt{\frac{1 + \gamma_t(x_t)}{1 - \gamma_t(x_t)}} + \delta_{max}\beta_t^{\frac{1}{2}}\gamma_t([x^*]_t)\sqrt{\frac{1 - \gamma_t([x^*]_t)}{1 + \gamma_t([x^*]_t)}} + 1/t^2.
\end{aligned}
\tag{41}
$$

The last two steps are based on the definition of $\gamma_t$.

It can be shown that

$$
\gamma_t^2(x_t)\frac{1 + \gamma_t(x_t)}{1 - \gamma_t(x_t)} \leq -C_{\delta_{min}}\log(1 - \gamma_t^2(x_t))
\tag{42}
$$

holds for some constant $C_{\delta_{min}} > 1$ related to $\delta_{min}$ when $0 \leq \gamma_t(x) \leq \frac{\sigma_t(x)}{\sigma_t(x)+\delta_{min}} < 1, \forall x$, and

$$
\gamma_t^2([x^*]_t)\frac{1 - \gamma_t([x^*]_t)}{1 + \gamma_t([x^*]_t)} \leq -\log(1 - \gamma_t^2([x^*]_t)).
\tag{43}
$$

Therefore,

$$
\sum_{t=1}^{T}\delta_{max}^2\beta_t\gamma_t(x_t)^2\frac{1 + \gamma_t(x_t)}{1 - \gamma_t(x_t)} \leq -C_{\delta_{min}}\delta_{max}^2\beta_T\sum_{t=1}^{T}\log(1 - \gamma_t^2(x_t))
\tag{44}
$$

$$
= -2C_{\delta_{min}}\delta_{max}^2\beta_T\sum_{t=1}^{T}\frac{1}{2}\log(1 - \gamma_t^2(x_t))
\tag{45}
$$

$$
= 2C_{\delta_{min}}\delta_{max}^2\beta_T I(f; F_T^a)
\tag{46}
$$

$$
\leq 2C_{\delta_{min}}\delta_{max}^2\beta_T I_T^{max}.
\tag{47}
$$

Similarly,

$$
\sum_{t=1}^{T}\delta_{max}^2\beta_t\gamma_t^2([x^*]_t)\frac{1 - \gamma_t([x^*]_t)}{1 + \gamma_t([x^*]_t)} \leq -\delta_{max}^2\beta_T\sum_{t=1}^{T}\log(1 - \gamma_t^2([x^*]_t))
\tag{48}
$$

$$
= -2\delta_{max}^2\beta_T\sum_{t=1}^{T}\frac{1}{2}\log(1 - \gamma_t^2([x^*]_t))
\tag{49}
$$

$$
= 2\delta_{max}^2\beta_T I(f; [F_T^a]^*)
\tag{50}
$$

$$
\leq 2\delta_{max}^2\beta_T I_T^{max},
\tag{51}
$$

where $[F_T^a]^* = [f^a([x^*]_1), \ldots, f^a([x^*]_T)]$.

By Cauchy-Schwarz inequality:

$$
\sum_{t=1}^{T}\delta_{max}\beta_t^{\frac{1}{2}}\gamma_t(x_t)\sqrt{\frac{1 + \gamma_t(x_t)}{1 - \gamma_t(x_t)}} \leq \sqrt{2C_{\delta_{min}}\delta_{max}^2\beta_T I_T^{max}T},
\tag{52}
$$

and

$$\sum_{t=1}^{T} \delta_{max} \beta_t^{\frac{1}{2}} \gamma_t([x^*]_t) \sqrt{\frac{1 - \gamma_t([x^*]_t)}{1 + \gamma_t([x^*]_t)}} \leq \sqrt{2\delta_{max}^2 \beta_T I_T^{max} T}. \tag{53}$$

Now we have:

$$R_T = \sum r_t \leq \sqrt{2C_{\delta_{min}} \delta_{max}^2 \beta_T I_T^{max} T} + \sqrt{2\delta_{max}^2 \beta_T I_T^{max} T} + \frac{\pi^2}{6}, \tag{54}$$

because $\sum_{t=1}^{\infty} 1/t^2 = \frac{\pi^2}{6}$. $\qquad\square$

**Notes on Equations** (42) **and** (43)**:** Consider $g(x) = x^2 \frac{1+x}{1-x} + c \log(1-x^2)$: $\frac{dg(x)}{dx} = -\frac{2x^2}{(1-x)^2(1+x)}(x^3 - (2+c)x - 1 + c)$.

Given any $x_R \in (0, 1)$, there exists a constant $c_{x_R} > 1$ related to the choice of $x_R$ such that $\frac{dg(x)}{dx} \leq 0, \forall x \in (0, x_R)$, by the nature of the cubic function $x^3 - (2 + c_{x_R})x - 1 + c_{x_R}$. Therefore, $g(x) \leq g(0) = 0, \forall x \in (0, x_R)$ when $c = c_R$.

Similarly, consider $h(x) = x^2 \frac{1-x}{1+x} + \log(1 - x^2)$, $\frac{dh(x)}{dx} = \frac{2x^2}{(1-x)(1+x)^2}(x^3 - 3)$. We have $\frac{dh(x)}{dx} \leq 0, \forall x \in (0, 1)$. So $h(x) \leq h(0) = 0, \forall x \in (0, 1)$.

It is now clear that Equations (42) and (43) hold, by replacing $x$ with $\gamma_t$.

# I  EXPERIMENTAL DETAILS

All the experiments are performed on Intel Xeon 8352Y processor with 256 GB memory. The benchmark functions are defined to be:

## I.1  HARTMANN

### I.1.1  Ground-truth objective function

It is a 6-dimensional function

$$f(x) = -\sum_{i=1}^{4} \alpha_i \exp\left(-\sum_{j=1}^{6} A_{ij}(x_j - P_{ij})^2\right), \tag{55}$$

where $x = [x_1, \dots, x_6]^T$, $\alpha = (1.0, 1.2, 3.0, 3.2)^T$, $A = \begin{pmatrix} 10 & 3 & 17 & 3.5 & 1.7 & 8 \\ 0.05 & 10 & 17 & 0.1 & 8 & 14 \\ 3 & 3.5 & 1.7 & 10 & 17 & 8 \\ 17 & 8 & 0.05 & 10 & 0.1 & 14 \end{pmatrix}$, $P =$

$10^{-4} \begin{pmatrix} 1312 & 1696 & 5569 & 124 & 8283 & 5886 \\ 2329 & 4135 & 8307 & 3736 & 1004 & 9991 \\ 2348 & 1451 & 3522 & 2883 & 3047 & 6650 \\ 4047 & 8228 & 8732 & 5743 & 1091 & 381 \end{pmatrix}$.

### I.1.2  Approximation Models

We consider two approximation models with the same evaluation cost of 1:

$$f^1(x) = f(x) + ([0.5, 0.5, 0.5, 0, 0, 0]x)S; \tag{56}$$

$$f^2(x) = f(x) + ([-0.5, -0.5, -0.5, 0, 0, 0]x + 1)S. \tag{57}$$

### I.1.3 Surrogate modeling

The ground-truth objective function is modeled with a GP with the RBF kernel

$$k(x_1, x_2) = \sigma \exp(\frac{||x_1 - x_2||_2^2}{l^2}). \tag{58}$$

The prior of the parameters are set to $\sigma \sim \mathtt{Unif}(1, 2)$, $l \sim \mathtt{Unif}(0.01, 0.5)$, where $\mathtt{Unif}(\mathtt{a}, \mathtt{b})$ represents a uniform distribution from $a$ to $b$. The input-dependent noise is modeled linearly with the prior of weights set to $\mathtt{Unif}(0, 1)$ and the prior of biases set to $\mathtt{Unif}(-1, 1)$. The number of samples to approximate the posterior is 64.

The implementation of MF-GP-UCB is set with the default configurations.

## I.2 BRANIN

### I.2.1 Ground-truth objective function

It is a 2-dimensional function

$$f(x) = a(x_2 - bx_1^2 + cx_1 - r)^2 + s(1 - t)\cos(x_1) + s, \tag{59}$$

where $x = [x_1, x_2]^T$, $a = 1$, $b = 5.1/(4\pi^2)$, $c = 5/\pi$, $r = 6$, $s = 10$, and $t = 1/(8\pi)$.

### I.2.2 Approximation Models

We consider two approximation models with the same cost of 1:

$$f^1(x) = f(x) + ([3.33, 3.33]x + 16.67)S; \tag{60}$$

$$f^2(x) = f(x) + ([-3.33, -3.33]x + 83.33)S. \tag{61}$$

### I.2.3 Surrogate modeling

The ground-truth objective function is modeled with a GP with the RBF kernel (58). The prior of the parameters are set to $\sigma \sim \mathtt{Unif}(100, 200)$, $l \sim \mathtt{Unif}(0.15, 0.75)$. The input-dependent noise is modeled linearly with the prior of weights set to $\mathtt{Unif}(0, 100)$ and the prior of biases set to $\mathtt{Unif}(-100, 100)$. The number of samples to approximate the posterior is 64.

The implementation of MF-GP-UCB is set with the default configurations.

## I.3 LEVY

### I.3.1 Ground-truth objective function

It is a 3-dimensional function

$$f(x) = \sin^2(\pi\omega_1) + \sum_{i=1}^{d-1}(\omega_i - 1)^2[1 + 10sin^2(\pi\omega_i + 1)] + (\omega_d - 1)^2[1 + \sin^2(2\pi\omega_d)], \tag{62}$$

where $d = 3$, $x = [x_1, \ldots, x_3]^T$, $\omega_i = 1 + \frac{x_i - 1}{4}$, for all $i = 1, \ldots, d$.

### I.3.2 Approximation Models

We consider two approximation models with the same cost of 1:

$$f^1(x) = f(x) + ([1, 1, 0]x + 20)S; \tag{63}$$

$$f^2(x) = f(x) + ([-1, -1, 0]x + 20)S. \tag{64}$$

### I.3.3 Surrogate Model

The ground-truth objective function is modeled with a GP with the RBF kernel (58). The prior of the parameters are set to $\sigma \sim \texttt{Unif}(40, 80)$, $l \sim \texttt{Unif}(0.2, 1)$. The input-dependent noise is modeled linearly with the prior of weights set to $\texttt{Unif}(0, 40)$ and the prior of biases set to $\texttt{Unif}(-40, 40)$. The number of samples to approximate the posterior is 64.

The implementation of MF-GP-UCB is set with the default configurations.

## I.4 MATERIALS DISCOVERY

The approximation model and ground-truth objective have been described in the main text.

The ground-truth band-gap objective is modeled with a GP with a RBF kernel (58), in which $\sigma = 4$, $l = 0.5$. The input-dependent noise is modeled with a GP with mean $m = 1$ and a RBF kernel parameterized by $\sigma = 0.5$, $l = 0.5$.

# J  ADDITIONAL DETAILS ON THE MATERIALS DISCOVERY PROBLEM (SEC 5.4)

## J.1  DISCUSSION ON DENSITY FUNCTIONAL THEORY (DFT) CALCULATION

Density Functional Theory (DFT) calculations are often used to predict a variety of material properties, including the band-gap [Jain et al., 2013]. These predictions can vary based on the structural and other configurations of the material. Despite their utility, DFT computations serve only as approximations to real experimental results, especially when calculating band-gaps. This limitation stems from the fact that DFT, strictly speaking, is a theory of the ground state of a material system, while the band-gap is essentially a property of the excited state. Consequently, for a single composition, the DFT-computed band-gaps can exhibit variation due to differences in material structural or other configurations. A case in point is the material CuBr, for which six different results are reported in the open-access Material Project (MP) dataset [Jain et al., 2013]. This highlights the challenges associated with accurately predicting band-gaps using DFT.

## J.2  DETAILS ON THE PRE-TRAINED MLP

The material compositions within the dataset incorporate a total of 80 distinct elements, with each material comprising two to four of these elements. Rather than using composition percentages directly as input for surrogate modeling, we initially generate 138 property-related features based on the material compositions. This is accomplished using the open-source Python package, matminer [Ward et al., 2018]. However, the data samples available in this generated 138-dimensional feature space are relatively sparse. To address this, we train a 3-layer Multilayer Perceptron (MLP) on the 1,019 samples that lack reported DFT-calculated band-gaps. The MLP accepts the 138-dimensional feature vector generated by matminer as input, with the two hidden layers comprising 8 and 2 neurons, respectively. The MLP is trained to minimize the mean square error between the MLP-predicted and ground-truth (experimental) band-gap values.

We then use the input of the MLP's final prediction layer as a feature extractor, distilling the original 138 features down to two. All MFBO methods are subsequently performed in this extracted 2-dimensional space. This approach effectively leverages the MLP as a tool for feature reduction, enhancing the explainability, scalability, and manageability of MFBO.