# OpenReview forum: "Multi-fidelity Bayesian Optimization with Multiple Information Sources of Input-dependent Fidelity"
_auai.org/UAI/2024/Conference — UAI 2024 poster_

### Official Review · Reviewer_w1uc · 2024-03-21

**Q2-1 Originality-Novelty:** 3
**Q2-2 Correctness-Technical Quality:** 3
**Q2-5 Clarity Of Writing:** 4

**Q1 Summary And Contributions:**

The authors "investigate MFBO when multi-fidelity approximations have input-dependent fidelity. By explicitly captur- ing input dependency for multi-fidelity queries in Gaussian Process (GP), our new input-dependent MFBO (iMFBO) with learnable noise models bet- ter captures the fidelity of each information source in an intuitive way.”

They try to "capture the varying fidelity by learning the input-dependent additive noise"

They also propose “a new acquisition function Noise-Variant Upper Confidential Bound (NVUCB) and theoretically derived a sub-linear regret bound.”

**Q2-3 Extent To Which Claims Are Supported By Evidence:**

4: Excellent: all claims are supported by very convincing evidence (in the form of comprehensive experimental evaluation, rigorous mathematical proofs, detailed (pseudo-)code, precise references, well-motivated and realistic assumptions) and the authors deliver what they promise.

**Q2-4 Reproducibility:**

3: Good: key resources (e.g. proofs, code, data) are available and key details (e.g. proofs, experimental setup) are sufficiently well-described for competent researchers to confidently reproduce the main results.

**Q3 Main Strengths:**

They paper tackles a worthwhile challenge untreated in the literature.

Their approach and execution is wells structured and complete.

They show deep understanding of the field via their detailed review of MFBO literature, smoothly leading into problem statement and solution.

Their empirical evidence study on real world materials discovery study is naturally challenging, but trade-offs are navigated relatively well.

**Q4 Main Weakness:**

I cannot identify any relevant weaknesses.

Ideally, they could provide one or two more real world tasks, but that's not necessary.

**Q5 Detailed Comments To The Authors:**

Questions:
a) Figure: f^1 and f^2 aren’t define/introduced well which can cause a lot of confusion
b) Fig3 c) red and blue dots are overlapping at times too much, maybe decrease opacity or something, not sure what’s a better way to plot that is: dots and crosses?
c) Can you do 30 or 40 iterations for Figure 4, not just 20? Helps with tightening confidence intervals hopefully


Typos
a) “This novel iMFBO enhances the sample efficiency of MFBO, enabling effective optimization of the elusive ground-truth objective function.” Seems somewhat redundant filler sentence, could be more concise

**Q9 Complying With Reviewing Instructions:**

Yes

---

> ### Author Rebuttal · Authors · 2024-04-04
>
> We are sincerely grateful for the reviewer's appreciation of this paper. For the three comments the reviewer gave,
>
> (a) We will introduce the $f^1$ and $f^2$ of Figure 1 in the updated version;
>
> (b) We updated the visualization of Figure 3(c) on the GitHub page https://github.com/TempAnonymous2023/iMFBO_rebuttal/blob/main/fid_choice.png by using circles and crosses to represent two types of queries. Hopefully, it could convey the idea more clearly;
>
> (c) We incorporate this suggestion to increase the iteration numbers and the suggestion from AKyU to add more information sources and perform a more complicated version of the band-gap maximization problem studied in Section 5.4 with a learned linear model as the third information source, considering 30 iterations of different MFBO methods. The results are illustrated on the GitHub page: https://github.com/TempAnonymous2023/iMFBO_rebuttal/blob/main/real_no_cost_addr.png. Our method still outperforms the benchmarks in this setting with three information sources and the claim in the main paper remains the same.

---

### Official Review · Reviewer_LJSt · 2024-03-22

**Q2-1 Originality-Novelty:** 2
**Q2-2 Correctness-Technical Quality:** 3
**Q2-5 Clarity Of Writing:** 2

**Q1 Summary And Contributions:**

The authors consider a problem where there are J noisy versions of a latent objective function such that their expectation corresponds to the objective function. The task is to find the optimum of the objective function by querying J functions. The key aspect of the problem is that J functions have heterogenous noise over the objective function. The authors propose a BO approach to solve this problem by modelling the joint surrogate over J functions and an acquisition function with theoretical guarantees (regret bounds).

**Q2-3 Extent To Which Claims Are Supported By Evidence:**

2: Fair: the main claims are somewhat supported by evidence (but the experimental evaluation may be weak, or does not match entirely with the claims, important baselines may be missing, proofs contain important ideas but lack rigor, algorithmic details are only discussed superficially, references are imprecise, assumptions are not sufficiently motivated or explicated, etc.).

**Q2-4 Reproducibility:**

3: Good: key resources (e.g. proofs, code, data) are available and key details (e.g. proofs, experimental setup) are sufficiently well-described for competent researchers to confidently reproduce the main results.

**Q3 Main Strengths:**

Given the setting the authors study, the method is principled and technically sound. Good empirical performance is demonstrated.

**Q4 Main Weakness:**

The paper does not study multi-fidelity BO. I disagree with the following sentence: "For multi-fidelity BO (MFBO), the ground-truth objective function f is not able to be directly queried or evaluated." In MFBO (Kandasamy et al., 2016; Takeno et al., 2020), the objective function (i.e. target fidelity) can be queried but with higher cost than its "approximations" (i.e. low-fidelities). The key challenge in MFBO is how to optimally balance between high accuracy but high cost and low accuracy but low cost functions. In the main paper it is also assumed that the cost is 1 across all fidelities so, in MFBO setting this would lead to a simple solution: always query the target fidelity.

Only linear noise model considered in the experiments? If so, why?

**Q5 Detailed Comments To The Authors:**

Please rewrite problem set-up part so that it is not misleading.

**Q9 Complying With Reviewing Instructions:**

Yes

---

> ### Author Rebuttal · Authors · 2024-04-04
>
> We appreciate the reviewer's time and effort reviewing this paper. We are grateful for the evaluation of our method and empirical results. We would appreciate that the reviewer could kindly check our response, which hopefully helps clarify the confusion and resolve the concerns.
>
> The reviewer's major concern is the positioning of this paper.
> We respect the reviewer's understanding of the terminology multi-fidelity Bayesian optimization (MFBO) and agree that the setup of this work is not exactly the same as the majority of the MFBO literature. However, we have tried our best to clearly describe our settings and formulations throughout the paper, to avoid misleading the readers. From the title "Multi-fidelity Bayesian Optimization with Multiple Information Sources of **Input-dependent Fidelity**", we explicitly point out the difference between our problem setup and traditional MFBO setups by stating that we are studying under **Input-dependent Fidelity**. We chose such a long title to try not to mislead the readers. In the related work section, we also discussed the contextualizing of our work by stating "To contextualize our contributions within the existing literature, we present the first-ever input-dependent MFBO (iMFBO) methodology that takes into account learnable input-dependent fidelity for queried evaluations facilitated by heteroscedastic learnable noise models". So that we don't unintentionally mislead any readers and to avoid potential confusion, we will incorporate your suggestion as well as that from reviewer AKyU to write a separate paragraph about the connection to and difference from the traditional MFBO in the related work section.
>
> To the best of our knowledge, this setup fits the best within the concept of MFBO because we are indeed considering Bayesian optimization with evaluation queries from different information sources that have different "fidelity" compared to the ground truth. The goal is to efficiently utilize the information from different sources to optimize the target function. With the same spirit as traditional MFBO, the only change in the setup of this work is the consideration of input-dependent fidelity, which is exactly the reason that our framework is novel and why we call the framework iMFBO. We consider the unit-cost scenario in the main text and the extension to the cost-aware version, where different information sources have different costs, in Appendix E with the corresponding algorithm and experimental results.
>
> We welcome any additional suggestions that may further improve the clarity of our paper and we would be more than happy to revise the abstract and introduction accordingly as needed.
>
> Regarding the sentence "For multi-fidelity BO (MFBO), the ground-truth objective function f is not able to be directly queried or evaluated", we agree with the reviewer that in many MFBO setups, the objective function (or objective function with observation noise) can be queried and viewed as the highest fidelity. MFBO may have various settings whose fidelity may represent different practical meanings. Although we are focusing on the case where every fidelity has a different noise level and (possibly) different cost for querying, motivated by many practical scenarios and also ample past work in MFBO, please note that our iMFBO framework can indeed handle the scenario mentioned by the reviewer -- i.e., when ground truth values may be queried at the highest cost, leading to noiseless evaluation for the "highest fidelity", and "lower fidelity" models having nonzero noise evaluations. We will revise the MFBO section accordingly to make the claims clearer so that the readers understand the main setting of our iMFBO work.
>
> For other points that were raised in the review comments, we think there may have been some misunderstanding, which we clarify below.
>
> In the experiments, we are **not** restricting the noise model to be linear models. We consider both the linear model (Section 5.3) and GP (Section 5.4) to model noise to demonstrate our "parameterized" and "non-parameterized" noise model settings introduced in Section 3.1. We apologize for not explicitly pointing this out and will explain this in the updated version.
>
> For reproducibility concerns, we want to note that the code is already available in the zip file in the Supplementary Material on the OpenReview page, right between the Abstract and Submission Number sections.
>
> Thank you again for the valuable feedback, which has been helpful in improving the clarity of our paper. We hope our rebuttal sufficiently addresses the reviewer's concerns, and we will make the aforementioned modifications in the revised manuscript accordingly to the reviewer's suggestions.

---

### Official Review · Reviewer_AKyU · 2024-03-27

**Q2-1 Originality-Novelty:** 3
**Q2-2 Correctness-Technical Quality:** 3
**Q2-5 Clarity Of Writing:** 3

**Q1 Summary And Contributions:**

The authors consider a specific instance of multi-fidelity Bayesian optimization, where the reliability of the different approximate evaluations are input dependent. The solution builds in part on existing components (heteroskedastic noise GPs) but includes also a new acquisition function and theoretical analysis. The method is demonstrated to work but only in simplified scenarios.

**Q2-3 Extent To Which Claims Are Supported By Evidence:**

2: Fair: the main claims are somewhat supported by evidence (but the experimental evaluation may be weak, or does not match entirely with the claims, important baselines may be missing, proofs contain important ideas but lack rigor, algorithmic details are only discussed superficially, references are imprecise, assumptions are not sufficiently motivated or explicated, etc.).

**Q2-4 Reproducibility:**

3: Good: key resources (e.g. proofs, code, data) are available and key details (e.g. proofs, experimental setup) are sufficiently well-described for competent researchers to confidently reproduce the main results.

**Q3 Main Strengths:**

- Simple and clear idea that matches well a specific scenario in multi-fidelity optimization. Like the authors say, ML approximations often are less accurate outside data-rich regions and we absolutely should be accounting for that when using them in BO. This paper provides the basic pattern on how it can be done, building on existing components.

- Clear artificial illustrations that help understanding how the model works.

- Solution builds on well-known components are is likely fairly robust and easy to use, but the work is not too straightforward; there is still a new acquisition function and some theoretical analysis

**Q4 Main Weakness:**

- Lack of convincing use case that really needs this tool. The problem described in Section 5.4 is nice, but I was disappointed on the authors not going any deeper. Since MLP is trivially fast to evaluate, the task is quite artificial -- there is not need to restrict the BO budget to making only a few queries of that approximation but we could just use all. The experiment is also otherwise a bit lazy -- why not consider e.g. multiple ML approximations?

- The relationship to majority of multi-fidelity BO literature remains in general a bit vague since you explicitly use unit cost throughout the text and experiments. Simply mentioning that the costs could be something else is not quite enough, but you would need to also explain how the weights would be accounted for (e.g. on the level of citing previous approaches and justifying how they could be easily incorporated) and ideally should include at least one experiment where the approximations have different costs (Section 5.4 is one, but you did not run it that way)

- The artificial data examples are also a bit too simplified from multi-fidelity perspective. While the benchmarks in Section 5.3 do show empirical gains, the setup where we can only evaluate two linear approximations is rather specific. Maybe you could here include also the option of evaluating the function itself but with a higher cost, and study a more common multi-fidelity setup?

**Q5 Detailed Comments To The Authors:**

The core idea is nice and appears to be novel, and it is well designed solution for a concrete problem. Overall the paper appears technically correct and the overall solution is sound, but there are some weaknesses in the problem formulation itself. The technical components of the solution are fairly straightforward; this is not a weakness as such, but the paper does not have additional strengths in contributing to the broader methodological literature.

The paper is otherwise well written and easy to read, but it is somewhat confusing to read due to positioning: You frame the contribution as part of the multi-fidelity BO literature but then consider only cases where all approximations are equally costly to evaluate, even in applications where this does not seem realistic. I would strongly encourage extending the discussion of this perspective to make the paper easier to digest. I personally think this is an interesting scenario and worthy contribution for the literature, but it could be presented better.

As explained above, my main problem with the paper is lack of very clear concrete example case and the simplified empirical experimentation. While the experiments do support the main claims, they do not lead to strong conclusions on practical value. The artificial data examples are illustrative, but the benchmark problems (Section 5.3) use only two very simple approximation models each (that are only explained in the Appendix) and the application scenario in Section 5.4 is also simplified and strictly speaking appears to be non-sensical since there is not need to restrict the number of function evaluations of the ML approximation since it is so fast to evaluate.

Questions: How easy it is to incorporate non-unit weighting for the approximations? How would your results change if you used realistic costs in the use-case of Section 5.4?

**Q9 Complying With Reviewing Instructions:**

Yes

---

> ### Author Rebuttal · Authors · 2024-04-04
>
> We thank the reviewer for the careful review and appreciate the constructive suggestions.
>
> One major concern from the reviewer is that the connection between this work and the major developments in MFBO is not discussed well and could lead to potential confusion. We thank the reviewer for pointing this out and will add a separate paragraph in the related work section to explicitly discuss the connection (we are considering BO with different evaluation or information sources that have different "fidelity" compared to the ground truth, and the goal is to efficiently utilize the information from different sources to optimize the target function) and the difference (we are not restricting to fixed high-low fidelity but consider input-dependent fidelity) to address this concern as well as reviewer LJSt's suggestion to reduce potential confusion.
>
> We would also like to note that we did discuss the extension considering the cost for different evaluations instead of only having the same weighting for the approximate evaluations in Appendix E, with an experiment whose results are illustrated in Figure 8. The cost-aware version of our acquisition function, MFNVUCBC, has the form of Equation 18. The results also demonstrate that the performance trends of different competing models are similar as reported in Section 5.4 when the
> querying costs are considered, with our MFNVUCB-based iMFBO outperforming other models as anticipated.
>
> Another concern from the reviewer is that we may need concrete realistic example cases that could show the potential of the proposed method in improving efficiency. We appreciate the suggestion and will try to identify such case studies. Based on the reviewer's suggestion to incorporate multiple ML algorithms, we performed a more complicated version of the band-gap maximization problem studied in Section 5.4 with a learned linear model as the third information source and a longer iteration of 30 based on reviewer w1uc's suggestion. The results are illustrated on the GitHub page: https://github.com/TempAnonymous2023/iMFBO_rebuttal/blob/main/real_no_cost_addr.png. Our method still outperforms the benchmarks in this setting with three information sources and the claim in the main paper remains the same. We would truly appreciate the suggestions from the reviewer if there are any other real-world benchmarks or additional experimental results that should be included.

---

### Meta-Review · Area_Chair_RQHf · 2024-04-15

The paper makes a solid contribution for the field by introducing a new variant of the problem setup and providing a robust method for addressing it, but suffers from lack of convincing demonstration of the problem's relevance. With such a demonstration it would be still stronger.